# Climate-warming-driven changes in the cryosphere and their impact on groundwater–surface-water interactions in the Heihe River basin

**Amanda Triplett and Laura E. Condon**

Hydrology and Atmospheric Sciences, The University of Arizona, Tucson, 85705, United States of America

**Correspondence:** Amanda Triplett (aktriplett@arizona.edu)

**Abstract.** The Heihe River basin in northwest China depends heavily on both anthropogenic and natural storage (e.g., surface reservoirs, rivers and groundwater) to support economic and environmental functions. The Qilian Mountain cryosphere in the upper basin is integral to recharging these storage supplies. It is well established that climate warming is driving major shifts in high-elevation water storage through loss of glaciers and permafrost. However, the impacts on groundwater–surface-water interactions and water supply in corresponding lower reaches are less clear. We built an integrated hydrologic model of the middle basin, where most water usage occurs, in order to explore the hydrologic response to the changing cryosphere. We simulate the watershed response to loss of glaciers (glacier scenario), advanced permafrost degradation (permafrost scenario), both of these changes simultaneously (combined scenario) and projected temperature increases in the middle basin (warming scenario) by altering streamflow inputs to the model to represent cryosphere-melting processes, as well as by increasing the temperature of the climate forcing data. Net losses to groundwater storage in the glacier scenario and net gains in the permafrost and combined scenarios show the potential of groundwater exchanges to mediate streamflow shifts. The result of the combined scenario also shows that permafrost degradation has more of an impact on the system than glacial loss. Seasonal differences in groundwater–surface-water partitioning are also evident. The glacier scenario has the highest fraction of groundwater in terms of streamflow in early spring. The permafrost and combined scenarios meanwhile have the highest fraction of streamflow infiltration in late spring and summer. The warming scenario raises the temperature of the combined scenario by $2\,°C$. This results in net groundwater storage loss, a reversal from the combined scenario. Large seasonal changes in evapotranspiration and stream network connectivity relative to the combined scenario show the potential for warming to overpower changes resulting from streamflow. Our results demonstrate the importance of understanding the entire system of groundwater–surface-water exchanges to assess water resources under changing climatic conditions. Ultimately, this analysis can be used to examine the cascading impact of climate change in the cryosphere on the resilience of water resources in arid basins downstream of mountain ranges globally.

## 1 Introduction

Mountains are an important source of freshwater for arid regions around the world (Qin et al., 2013; Viviroli and Weingartner, 2004; Wu et al., 2015). The cryosphere (i.e., water in mountainous, alpine regions stored as glaciers, snow, permafrost and rain) plays a critical role in moderating water availability to downstream watersheds (Gao et al., 2018). It temporally redistributes winter precipitation to higher-demand periods like the spring and summer (Viviroli et al., 2011) and reduces the variability of flow (Wang and Cheng, 2000).

High-latitude, cold regions have greater sensitivity to global warming (Chen et al., 2018; Jones and Rinehart, 2010; Zhang et al., 2020). The warming rate in the Tibetan Plateau, the largest and highest mountain range in the world, is twice the global rate (You et al., 2020). This accelerated warming of the cryosphere has substantially altered water cycles and streamflow (Chen et al., 2018; Wu et al., 2015; Xu et al., 2015; Zhang et al., 2016). Alterations in the quantity of cryosphere water storage and timing of discharge can

change downstream water availability and how it is allocated (Chen et al., 2018; Xu et al., 2015). However, the impact of cryosphere melting on downstream systems is not fully understood.

The Heihe River basin is an example of a system that has been impacted by the warming climate. It is a semi-arid, agriculturally important region located in northwest China (Fig. 1). The Qilian Mountain cryosphere in the upper basin is the region's primary water source (Wang and Cheng, 2000; Li et al., 2016). The movement of water from the high-precipitation upper reaches to the arid valley floor has been critical for downstream development (Y. Liu et al., 2019). It has allowed for the expansion of irrigated agriculture, which accounts for over 90 % of water usage in the middle basin (Chen et al., 2005; Deng and Zhao, 2015; Sun et al., 2016). However, this reliance on water from the upper reaches makes the middle basin more vulnerable to warming-induced changes in the cryosphere than other areas with higher local precipitation (Kang et al., 1999).

The upper basin is expected to undergo significant changes in glacier volume, permafrost coverage and precipitation due to climate change. Future projections for northern Asia, where the Heihe River basin is located, indicate that precipitation will likely increase (Shi et al., 2006; Zhang et al., 2016). However, estimates for the timing and volume of future precipitation in high mountain areas are variable (IPCC, 2014). Increasing warming trends, on the other hand, are essentially certain (IPCC, 2014). Thus, in this study, we focus on processes resulting from increased temperature alone, such as glacial melt and permafrost degradation.

Glacial contribution to streamflow is of particularly high importance in arid basins (Viviroli et al., 2011). Glaciers can stabilize flows, especially during hot or dry years (Chen et al., 2015; Qin et al., 2014). The ability of glaciers to buffer streamflow depends on glacial volume, melt rate and the balance with evapotranspiration (ET). Under climate warming, it is estimated that glaciers in the upper basin may disappear entirely by the middle of the 21st century (Chen et al., 2018; Wu et al., 2015). In this case, the glacial contribution to flow and its moderating effect in warmer months will eventually vanish.

Warming temperatures have also caused significant permafrost degradation in alpine regions around the world, including the Qilian Mountain cryosphere (Gao et al., 2018; Ma et al., 2019; Niu et al., 2010; Song et al., 2019). Permafrost acts as an impermeable boundary to infiltrating water. For this reason, permafrost-dominated catchments tend to have higher peak flows and lower baseflows, with primarily short, lateral groundwater flow paths (Carey and Woo, 2001; Niu et al., 2010; Ye et al., 2009). When permafrost degrades, hydraulic conductivity increases, and water can infiltrate to deeper depths and take longer flow paths (Ma et al., 2019; Niu et al., 2010). This results in lower peak flows as more water infiltrates instead of running off and higher baseflows as more groundwater enters streams (Carey and Woo, 2001;

Ma et al., 2019). There is also an increase in the volume of ground-ice meltwater, which is the release of water stored as ice within permafrost (Ma et al., 2019).

Many studies have examined the contribution of glacial meltwater to streamflow in the upper Heihe River basin. These estimates range from 3 % up to about 10 % for the Heihe River (Chen et al., 2018, 2015; Gao et al., 2018; Li et al., 2018; Niu et al., 2010; Wu et al., 2015). Cryosphere meltwater contribution to smaller rivers, like the Hulugou (Qilian Mountains), may be as high as 32 % (Li et al., 2014). This contribution only occurs during the thawing season, which is from April to October (Gao et al., 2018). There is not full agreement on how much these glaciers contribute to total flow and what streamflow may look like after they disappear.

Previous work has also quantified the impact of permafrost degradation on streamflow. Increasing winter streamflow trends in alpine regions can be attributed to permafrost degradation processes as there are very few alternate sources of water at this time (Gao et al., 2018; Ma et al., 2019; Niu et al., 2010). This increase is often only significant in basins with initially high permafrost coverage (Niu et al., 2010; Ye et al., 2009), such as the Heihe River basin. The estimated increase in runoff in the freezing season from permafrost degradation in the upper Heihe River basin is around 50 % from 1971 to 2010, associated with an 8.8 % loss in permafrost area (Gao et al., 2018). While the change in flow is measured during the freezing season, degrading permafrost could impact baseflow in all seasons (Jones and Rinehart, 2010; Walvoord and Striegl, 2007).

Numerical, process-based hydrologic models have been used previously to study the Heihe River basin. Cryosphere response to global warming in the upper basin was studied by Chen et al. (2018) and Gao et al. (2018). Models have also been used to examine a wide range of water resource issues in the middle and lower reaches of the Heihe River basin. For example, the model HEIFLOW (Hydrological-Ecological Integrated Watershed-scale Flow) has been used to simulate groundwater–surface-water interactions, agricultural operations, ecohydrological response and reservoir impacts amongst other topics (Han et al., 2021; Li et al., 2018; Sun et al., 2018; Tian et al., 2018; Tian et al., 2015a, b; Yao et al., 2018; Yao et al., 2015a, b). However, to our knowledge, no studies have examined the impact of changes in upper-basin streamflow due to cryosphere processes on both groundwater and surface water in the middle basin.

We address this gap by modeling the middle-basin response to cryosphere changes using the integrated hydrologic model ParFlow-CLM (PARallel FLOW-Common Land Model). ParFlow-CLM is designed to capture interactions between groundwater, surface water and land surface fluxes (Jones and Woodward, 2001; Kollet and Maxwell, 2006; Maxwell et al., 2015). It is thus well suited to examine evolving watershed dynamics. Using this approach, we explore how groundwater–surface-water interactions and water storage in the middle basin evolve as a result of changing stream-

flow coming from the cryosphere, how these processes vary seasonally, and how projected warming in the middle basin can shift this response.

## 2   Data and methods

### 2.1   The study area

The Heihe River basin is a semi-arid catchment with an area of approximately 130 000 km² in northwest China (Fig. 1). It is located in the Hexi Corridor, one of the most arid regions in the world (Lu et al., 2015). The basin decreases in elevation and increases in temperature and aridity moving from south to north. Elevation varies from about 5600 to 900 m (Yao et al., 2018), long-term average temperature ranges from −4 to 10 °C (Y. Liu et al., 2019), precipitation ranges from 800 mm to below 50 mm (Y. Liu et al., 2019), and potential ET ranges from 700 mm (L. Zhang et al., 2015) up to 2300 mm (Deng and Zhao, 2015).

The Heihe River basin has three principal sections: the upper, middle and lower basin. The upper basin is located on the northern edge of the Tibetan Plateau and contains the Qilian Mountains and the headwaters of the Heihe River, the largest river in the basin. This is the primary runoff generation area for the rest of the basin, contributing about 70 % of total river runoff in the lower reaches (Y. Liu et al., 2019; Yang et al., 2014). The middle Heihe is a flat oasis area where most human settlement and economic activity is located. The middle basin uses an estimated 80 %–95 % of the available freshwater (Deng and Zhao, 2015; Liu et al., 2009; Sun et al., 2016; Yang et al., 2015; Yao et al., 2015b). Of this, 80 %–90 % is consumed by irrigated agriculture (Chen et al., 2005). The lower basin is primarily the Gobi Desert and has little human development. It contains the two terminal lakes of the Heihe River.

### 2.2   Hydrologic modeling approach

We elected to use the integrated hydrologic model ParFlow-CLM in this study. ParFlow-CLM is a fully integrated hydrologic modeling platform that simulates surface and subsurface processes together. It has been used extensively in hydrologic studies of groundwater–surface-water interactions, the food–energy–water nexus and climate change in small- to large-sized basins, including the entire continental US (Condon et al., 2020; Condon and Maxwell, 2014; Ferguson and Maxwell, 2010; Hein et al., 2019). ParFlow-CLM has also been used to model the Central Valley in California, a semi-arid, mountain–valley agriculture system with many parallels to the Heihe River basin (Gilbert and Maxwell, 2017, 2018; Thatch et al., 2020). In the subsurface, variably saturated flow is solved using the mixed form of Richards' equation. Overland flow is calculated by solving the kinematic wave approximation and Manning's equation (Kollet and Maxwell, 2006). Further details about the workings of

ParFlow are provided in Ashby and Falgout (1996), Jones and Woodward (2001), Kollet and Maxwell (2006), Maxwell et al. (2015), and Maxwell (2013). ParFlow is coupled to the Common Land Model (CLM). CLM is a land surface model which handles the surface-water–energy balance (Maxwell and Miller, 2005; Kollet and Maxwell, 2008).

There has been previous hydrologic model development in the Heihe basin. HEIFLOW is a GSFLOW (Coupled Ground-Water and Surface-Water Flow Model)-based model (Markstrom et al., 2008). GSFLOW was developed by the USGS and couples MODFLOW (Modular Ground-Water Flow Model) and PRMS (Precipitation-Runoff Modeling System). The vadose zone, rivers, lakes and other components are defined ahead of time and handled by coupled packages. GSFLOW was enhanced across several studies to include modules which handle surface diversion and pumping (Tian et al., 2018; Tian et al., 2015a, b), dynamic vegetation growth (Sun et al., 2018), and subgrid parameterization of soil and irrigation water (Han et al., 2021).

The primary difference between the ParFlow-CLM model we present here and HEIFLOW is that ParFlow solves variably saturated flow in all subsurface cells. Additionally, overland flow is fully integrated with the subsurface in ParFlow through a free-surface overland flow boundary condition that allows rivers to form and disappear as moisture changes (Kollet and Maxwell, 2006). The approach used by ParFlow means that there is no need for a priori specification of saturated zone, vadose zone, river network, etc. prior to simulation. This approach allows for a dynamic evolution of groundwater–surface-water interactions and accurate accounting of exchanges of water between surface and subsurface layers. This capability is of importance in modeling the middle Heihe because of the high rate of conversion between surface water and groundwater (Yao et al., 2015a; Wang and Cheng, 2000).

### 2.3   Model inputs

Details about the data used in the model can be found in Table 1. The reference corresponding to the data source column will provide a DOI to the original data or the publication which details its use. For example, some of the data underwent pre-processing and parameterization to be used in the construction of the HEIFLOW model, for which details can be found in Tian et al. (2018), Tian et al. (2015a, b) and Yao et al. (2015a, b). These data served as our source data and will be referred to as such throughout the paper.

ParFlow-CLM requires gridded inputs for hydraulic conductivity ($K$), specific storage and porosity. The source data for $K$ were parameterized in Tian et al. (2015b), resulting in 92 unique values ranging from 0.001 to 5.625 m h⁻¹. We aggregated these values for our study into 15 soil and 15 geological units to facilitate calibration by hydrogeologic group. We also assigned a value of 0.001 m h⁻¹ to regions in the vertical domain of the ParFlow-CLM model that had no source

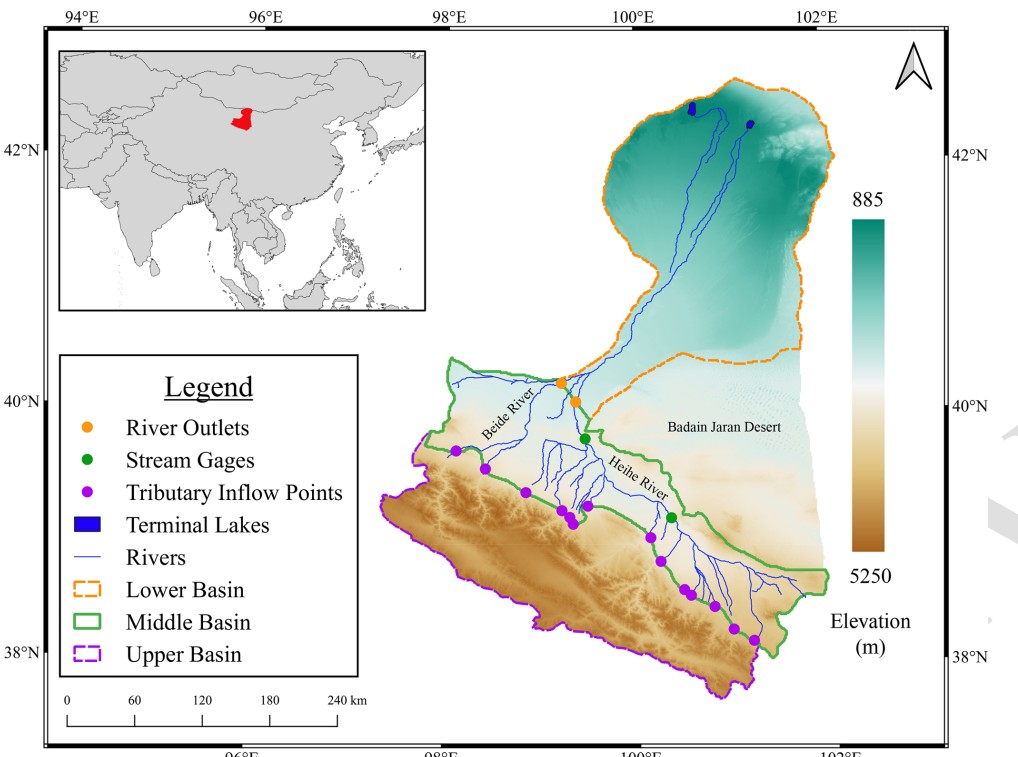

**Figure 1.** The location of the Heihe River basin in red in the inset to provide its geographic context in Asia. The upper basin is outlined in purple, the middle basin in green and the lower basin in orange. The Badain Jaran Desert is labeled and is hydrologically connected to the basin. The gaged inlets between the upper and middle basin are in purple. The two river gages on the main stem of the Heihe River are in green and were used for calibration. The locations of the Beide and Heihe River outlets are shown in orange. The two terminal lakes are colored in blue at the end of the river network. Elevation ranges from ~ 5250 to 885 m and is shown in the legend to the right.

**Table 1.** Source data.

| Starting variable | Data source | Original units | Spatial resolution | Time of data | Model input |
|---|---|---|---|---|---|
| Geolayers | Yao et al. (2015a) | m | 1 km × 1 km | 2000 | Vertical discretization |
| DEM | HPDMC | m | 1 km × 1 km | 2008 | $X/Y$ slopes |
| Hydraulic conductivity ($K$) | Tian et al. (2015b) | $\mathrm{m\,d^{-1}}$ | 1 km × 1 km | 2000 | $K$ |
| Specific storage (SS) | Tian et al. (2015b) | $\mathrm{m^{-1}}$ | 1 km × 1 km | 2000 | SS |
| Specific yield (SY) | Tian et al. (2015b) | [] | 1 km × 1 km | 2000 | Porosity ($n$) |
| Groundwater boundary condition (GWBC) | Tian et al. (2015b) | $\mathrm{m\,d^{-1}}$ | Boundary points | Average from 2000–2012 annual Data | GWBC |
| Surface water boundary condition (SWBC) | HPDMC | $\mathrm{m^3\,s^{-1}}$ | 14 stations | 2000–2012 (daily and/or monthly) | SWBC |
| Land cover | HPDMC | NLUD-C groups | 1 km × 1 km | 2011 | Land cover, Manning values |

data. The intention is for this region to be considered as bedrock.

The source data had information regarding specific yield but not the required variable, porosity. A majority of the domain acts as an unconfined aquifer, with only locally confining conditions (Yao et al., 2015a, b). This was confirmed by the source data, where the difference in conductivity be-

tween aquitard and aquifer layers was not noticeable across much of the domain and was consistently much less than the decrease in conductivity with depth. For our model, we assume that specific yield can be used as a reasonable estimate for porosity. This is of course an assumption, but it is likely that the difference between specific yield and porosity is less than the considerable spatial uncertainty in specific yield. We

also simplified the values in the specific yield data from 17 unique values ranging from 0.05–0.35, as calibrated in Tian et al. (2015b), to three intervals of 0.1, 0.2 and 0.3 and 0.05 for bedrock. A lumped approach allowed for greater focus in the calibration process on variables the model was more sensitive to, such as $K$.

The source data used two unique values for specific storage and were assigned by hydrogeologic unit. The spatial distribution of these values did not vary with depth, and we applied the data unchanged to our model layers. We categorized the porosity and specific storage variables into seven groups, six for each unique pair of porosity and specific storage values and a seventh group representing bedrock for regions in our vertical domain that had no source data. This group was assigned the lowest value from the source data for each respective variable. This corresponds to 0.05 for porosity and $1.0^{-4}\,\text{m}^{-1}$ for specific storage. These values are corroborated by the literature as reasonable values for bedrock (Huntington and Niswonger, 2012).

The digital elevation model (DEM) (Table 1) was processed to ensure adequate surface drainage of every cell in the domain to either the stream network or domain boundary. The process was accomplished using PriorityFlow, an open-source R package which is a modified priority flood and global-slope enforcement algorithm (Condon and Maxwell, 2019). The result is a smoothed, fully draining DEM, which was used to produce the $X$ and $Y$ slope files which are required input for ParFlow-CLM. The processed DEM was also used to calculate drainage areas and stream orders which were later used to define Manning's roughness parameters.

The land cover dataset uses the NLUD-C (national land use/cover database of China) classification scheme for 2011 (Table 1). Land cover has not been static over the period of simulation. For example, farmland, forest and built-up land have all increased due to the expansion of agriculture and other economic activity in the basin, while grasslands, waterbodies, wetlands and desert have all decreased, likely converted to the previous land types (Hu et al., 2015). However, land conversion slowed considerably after the year 2000, and most natural oases in the basin had already been converted to farmland by 1975 (Lu et al., 2015). In addition, future land use patterns are not expected to be appreciably different from the year 2000 (L. Zhang et al., 2015). For these reasons, we made the decision to use the 2011 land cover map for our simulations. The land cover map was converted from NLUD-C to the IGBP (International Geosphere-Biosphere Programme) classification as that is the categorization required by CLM. NLUD-C categories were matched with the closest IGBP group based on descriptions. In some cases, CLM parameters such as leaf area index (LAI) or canopy height were altered to better match with the NLUD-C land cover categories. The result is an 18-category IGBP land cover map that matches the 2011 NLUD-C map. Finally, the IGBP land cover map and the stream order map of the domain produced by the topographic-processing workflow

were used to create a spatially variable Manning's roughness value grid. The conversions for land cover type and stream order to Manning's roughness were obtained from Foster and Maxwell (2018) and the 2015 WRF-Hydro User Guide version 3.0 (Gochis et al., 2015).

The climate forcing variables required to run CLM are long- and shortwave radiation, precipitation, atmospheric pressure, specific humidity, and $u$ and $v$ wind components. The input climate dataset used is CMFD (China Meteorological Forcing Dataset) detailed in He et al. (2020). It has a temporal resolution of 3 h and a spatial resolution of 0.1° or $\sim 10\,\text{km}$. Although there are several other climate forcing datasets available, this one was selected as it had almost all the variables required to run CLM, was available for the entire simulation period and had good spatial and temporal resolutions. In order to fit the data to our 1 km modeling grid, the climate data from CMFD were extracted and resampled. Then, the 3 h time step was divided into 1 h time steps, where each span of 1 h contains the same average data. The CMFD data only contained total wind as opposed to the $u$ and $v$ wind vectors required by CLM. To address this issue, we used wind direction data generated by a high-resolution regional climate model specifically designed for the Heihe River basin (Xiong and Yan, 2013). This was used to derive the wind direction angle, which was then applied to the CMFD wind magnitude to obtain $u$ and $v$ wind components.

## 2.4 Natural-flow-state modeling

Our simulations focus on the natural state of the Heihe (i.e., ignoring anthropogenic activities such as groundwater pumping and irrigation). Natural-flow models have often been used to isolate anthropogenic contributions to flow regime changes, quantify water available to managers and regulators, and study catchment response to climatic change, even in heavily managed systems (Terrier et al., 2020). Following this approach, we exclude surface water diversion and groundwater-pumping processes from our simulations. These water uses are significant in the Heihe River basin (Li et al., 2018; Tian et al., 2018). However, the complexity of addressing climate and water use change simultaneously can make it difficult to assess purely climatic impacts (Terrier et al., 2020). As the broader goal of this paper is to establish trends in water availability in the Heihe in response to future climate warming, a natural-flow-state model was deemed appropriate.

The inflow to the middle Heihe (our study domain) is almost entirely natural as there is very little water use upstream. However, the stream gages and wells inside the domain are within areas which have been heavily impacted by surface diversion and pumping for decades (Li et al., 2018; Lu et al., 2015; Tian et al., 2018). To correct for this, we apply a naturalization method to compare model and observed streamflow time series. We used a water balance method outlined in A. Zhang et al. (2015), which assumes that water lost

between an upstream and downstream gage roughly corresponds to the water diverted between stations. We quantified the flow lost between the inlet and the HRB2 (Heihe River basin) gage (Fig. 1, closest to outlet) and added that quantity of water back to the streamflow time series for HRB2. It should be noted that this is an approximation as this method does not take into account the delayed and indirect impact groundwater withdrawals have on river flows (Terrier et al., 2020). Additionally, other tributaries which connect to the Heihe River inside the domain are ungaged. This means that, although the flows in these tributaries are also impacted by diversion and pumping, we cannot adequately correct for them. Thus, despite naturalization, we still expect the flows in our model to be higher than the observed data. For this reason, we focused on matching winter flows when there is little pumping or diversion, as well as the timing of flow rather than magnitude.

## 2.5 Model configuration and initialization

The modeling domain selected is the middle Heihe, as shown in Fig. 1. The domain has a spatial resolution of 1 km. This is the resolution of most of the source data (Table 1). The domain dimensions are 360 (horizontal) by 270 (vertical) cells. The domain was divided into 14 vertical layers of varying thickness (note there is no lateral variation in thickness) as follows: 0.1, 0.3, 0.6, 1.0, 10.0, 10.0, 30.0, 30.0, 30.0, 30.0, 30.0, 100.0, 100.0 and 100.0 m. The top four layers correspond to soil and the bottom 10 to geologic layers. This results in a total depth of 472 m. The thickness of the bottom 10 layers was selected to capture the variability seen in the hydrogeologic data of the basin. The HEIFLOW model had five vertical layers that varied in thickness laterally and corresponded to the shallow unconfined aquifer, the first aquitard, the shallow confined aquifer, the second aquitard and the deep confined aquifer (Yao et al., 2015b), with a maximum depth of 2094 m (Yao et al., 2015a; Tian et al., 2015b). However, a no-flow boundary was imposed in the model across most of the domain at a much shallower depth. For example, only 8 % of the domain contains data past a depth of 1000 m. We selected a thickness of 472 m for our model as it retains most of the spatial variability of the input data, allows for the resolution of groundwater flow paths on the time order of simulation (11 years) and maximizes model performance.

We applied a constant flux boundary condition along the border between the upper and middle Heihe and no-flow boundaries along the rest of the subsurface. The flux across the boundary with the upper basin was calibrated for use in the HEIFLOW model (Tian et al., 2015b; Tian et al., 2018) (Table 1). We adjusted this flux for our modeling domain by subtracting the flow entering below our model depth of 472 m. The remaining flux was applied evenly to all non-bedrock cells along the boundary (i.e., $K$ greater than 0.004 m h$^{-1}$) (Gleeson et al., 2014; Huntington and Niswonger, 2012). This resulted in a flux value of $1.7^{-4}$ m h$^{-1}$ that was applied to all non-bedrock cells on the southern boundary. While groundwater fluxes are likely to change seasonally, there were no available data to support intra-annual values. Additionally, the groundwater flux only makes up approximately 5 % of the average annual water input to the model. Further, during calibration, values of +75 % and −75 % of the original values were tested and had minimal impact on model output. As seasonal variation is unlikely to fall outside these values, we determined that a constant boundary condition was sufficient.

There are 14 gaged rivers entering the middle basin from the upper basin (Fig. 1). We injected water into the model according to the flow at these gages. While many of the stream gages have daily data, others only have monthly data (Table 1). In cases where daily data for a gage do not exist, the daily fraction of monthly flow was calculated for the closest gage with daily data. These fractions were then multiplied by monthly flow to interpolate daily data for the target stream gage. Based on annual averages for streamflow, precipitation minus ET from the climate forcing and groundwater influx from the boundary condition, the input-water breakdown for the middle Heihe domain is approximately 75 % from streamflow, 20 % from precipitation and 5 % from the groundwater boundary condition.

In order to accurately calibrate the model, an appropriate groundwater configuration was needed to serve as an initial pressure state. To do this, the model was run with a long-term recharge forcing at the land surface until groundwater storage as a percent of recharge was less than 1 %. The long-term recharge forcing was derived from the average difference between precipitation and ET from the climate forcing data. This resulting pressure state was used as the starting point for all parameter calibrations. For model calibration, streamflow observations from the 2011 WY (water year) at two gages, HRB1 and HRB2, as well as average water table depth (WTD) at 44 groundwater observations wells, were used to assess model performance. Calibration was performed by manually adjusting the groundwater boundary condition, $K$, and Manning's roughness coefficients.

After final parameters were selected, a new model initialization was performed. This was done because groundwater systems move very slowly, and it was necessary to ensure that the model was once again in equilibrium before running the scenarios. During this final initialization, the model was first brought to steady state with the long-term recharge forcing and then with the climate forcing to bring the groundwater system as close as possible to the conditions at the start of the scenario period (2001 WY). We judged this as complete in both cases when subsurface storage change as a percent of recharge fell below 1 %. This final model pressure state was then used as the starting point for all scenarios outlined below.

## 2.6 Cryosphere melt scenarios

We designed five scenarios to model the middle-basin response to future climate change: (1) a baseline scenario, which uses historic climate and streamflow data to model an unaltered, natural flow state. Next, three scenarios explore the middle-Heihe response to changing streamflow input from the cryosphere as a result of warming: (2) a glacier scenario simulating the loss of the glacial contribution to streamflow; (3) a permafrost scenario capturing increases in baseflow as a result of permafrost degradation; and (4) a combined scenario, which models both the glacial and permafrost impacts on streamflow together. Finally, (5) a warming scenario captures temperature increase in the middle basin.

The above scenarios are modeled using input data from the 2001 to the 2011 WY. This period was selected for three reasons. The first is data availability. The second is that we are interested in changes in groundwater storage. As groundwater is slow moving, if we want to capture the longer-term trends in storage, it is important to simulate for as many years as possible. Last, the period is representative of wet, normal and dry years, which makes it ideal to examine climate impacts on hydrologic processes (Tian et al., 2018).

The baseline scenario represents observed historical conditions. Here, we apply daily streamflow from historic data at the 14 gage locations between the upper and middle basins (Fig. 1). Figure 2 shows idealized annual streamflow separated into components. For the baseline, all three components are applied at their historic fraction throughout the year. The light-blue precipitation component, which consists of rain and snowmelt, remains unaltered in all scenarios.

The glacier scenario is designed to represent a future in which the glaciers in the upper basin have completely disappeared. To do this, we remove the fraction of streamflow contributed by glacial melt during the thawing season (April to October). This is represented by the dark-blue component in Fig. 2. Freezing and thawing periods were taken from Gao et al. (2018). The contribution of glacial melt to upper-basin streamflow has been estimated by several studies to be ∼ 3 %–10 % (Chen, 2014; Chen et al., 2018; Gao et al., 2018; He et al., 2008; Li et al., 2016; Wu et al., 2015; Yang, 1991). Here, we use a slightly larger value of 15 %, which provides an upper bound on what is likely and considers uncertainties in prior estimates. For example, prior estimates are based on historic melt rates and cryosphere interactions and do not account for potential nonlinearity under future climate change. Also, most of these studies exclusively examine the Heihe River and neglect smaller tributaries that feed the lower basin and may have different glacial fractions (Li et al., 2014). By using a larger value for the glacial fraction, we can set a lower bound on future water supply in the region.

The permafrost scenario models changes in streamflow as a result of permafrost degradation. The baseflow component of streamflow, shown in orange in Fig. 2, is altered in this scenario. Gao et al. (2018) found that winter flow increased by

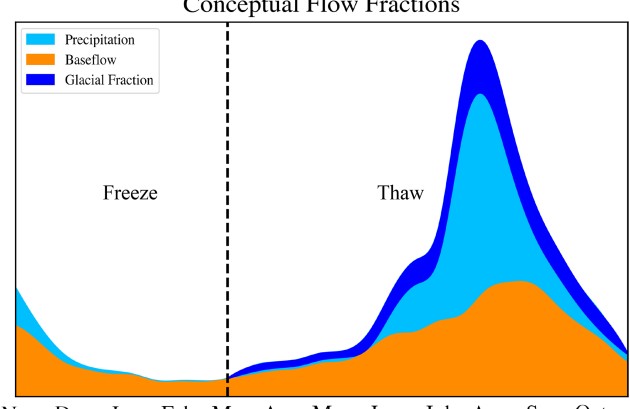

**Figure 2.** Conceptual model of streamflow with each of the flow components highlighted. Freeze and thaw refer to times of year where we expect water to melt in the cryosphere (thaw) or remain frozen (freeze). The precipitation fraction is light blue and consists of rain and snowmelt. This fraction is unaltered in the scenarios. The glacial fraction is dark blue. This fraction corresponds to 15 % in the scenarios and is removed from streamflow during the thawing season for the glacier, combined and warming scenarios. The orange fraction represents baseflow and is increased by 50 % in the permafrost, combined and warming scenarios.

∼ 50 % in the upper basin from 1970 to 2010. This increase in baseflow corresponded to an estimated 8.8 % reduction in permafrost area. We assumed a similar loss of permafrost area by the mid-century and chose to increase our baseflow by 50 % for the permafrost scenario. Gao et al. (2018) chose to assess the impact of permafrost degradation in the freezing season because this is when other contributions to flow are minimal and it is easier to isolate. However, we apply this increase year-round as opposed to only in the freezing or thawing season because subsurface permeability changes and enlargement of the groundwater reservoir due to permafrost degradation could impact baseflow year-round (Jones and Rinehart, 2010; Walvoord and Striegl, 2007). Although there are also likely reductions in peak flows in the thawing season due to permafrost degradation (Carey and Woo, 2001; Ma et al., 2019), it is difficult to generalize these impacts due to other contributions to streamflow, such as by precipitation. Thus, a reduction in peak flows would be arbitrary. The permafrost scenario serves as an upper bound for future water supply in the basin.

To apply the baseflow change, we performed baseflow separation on the observed streamflow using the digital-filtering method outlined in Z. Liu et al. (2019). Digital filtering separates high- from low-frequency signals, in this case runoff from baseflow. Equation (1) solves for surface runoff at the current time step ($Q_{dt}$) and Eq. (2) solves for baseflow ($Q_{bt}$). $\beta$ is the filtering parameter, and $T$ is the number of passes with the digital filter. The initial parameterization for $\beta$ and $T$ was taken from estimates for the upper Heihe basin

(Z. Liu et al., 2019; Zhao et al., 2016). After visual inspection, $\beta = 0.90$ and $T = 3$ were selected as the best fit for our data.

$$Q_{\mathrm{dt}} = \beta Q_{d(t-1)} + \frac{(1+\beta)}{2} \left[ Q_{\mathrm{t}} - Q_{(t-1)} \right] \qquad (1)$$

$$Q_{\mathrm{bt}} = Q_{\mathrm{t}} - Q_{\mathrm{dt}} \qquad (2)$$

The combined scenario represents a system where changes in flow due to glacial melt and permafrost degradation occur at the same time. For this case, we apply perturbations to streamflow that are identical in timing and volume to those made in both the glacier and permafrost scenarios (i.e., 15 % reduction in thawing season flow and 50 % increase in baseflow year-round). These changes are made according to the same reasoning as outlined in the descriptions of the glacier and permafrost scenarios above. In many ways, the combined scenario is the most realistic future representation of streamflow as we do expect glacial reductions and permafrost degradation to both occur. We performed the isolated glacial and permafrost cases in order to quantitatively isolate the different signatures that these changes have and to set upper and lower bounds on water supply.

The warming scenario is designed to evaluate the impact of future warming in the middle basin on the hydrologic system. The warming scenario is identical to the combined scenario except for a global increase of 2 °C in the CMFD temperature forcing data. We selected 2 °C as it is a reasonable mid-century estimate for global temperature increase (IPCC, 2014). In line with previous studies, we decided that simplifying the temperature increase would allow us to better isolate the hydrologic response to warming (Condon et al., 2020).

## 3 Results

Results are organized into four subsections. Section 3.1 outlines the performance of the baseline model with regards to streamflow and WTD observations. Section 3.2 covers all results related to streamflow specifically. This includes overall time series, anomalies from the baseline and seasonal patterns. Similar results are covered in Sect. 3.3 for subsurface storage. Section 3.4 contains spatial results which allow for the assessment of warming impact between the scenarios.

### 3.1 Baseline model performance

To assess model performance, we compared model streamflow to observed data at gage HRB2 on the Heihe River. HRB2 is the furthest downstream gage and the closest to the outlet (Fig. 1). The streamflow performance of the baseline scenario is shown in Fig. 3. It is important to note that we are modeling a natural flow state, as discussed in Sect. 2.4. Observational data are subject to operations like pumping and diversion. As we do not include these processes, we expect our model streamflow to be higher than observed. For this reason, our main targets were to match freezing-season baseflow (when there is little diversion) and streamflow timing. We do not expect to see an overall perfect match between model and observations.

Observed streamflow was naturalized according to the method outlined in A. Zhang et al. (2015). The difference between the observed and naturalized streamflow is illustrated in Fig. 3b and c. Figure 3b shows the model (blue) and observations (red) in the 2001 WY with no naturalization applied. The model matches flow well in the freezing season (November to March), as expected, due to minimal diversion, although the model still tends to overestimate, likely due to permanent differences in groundwater–surface-water interactions between a natural and managed catchment (Terrier et al., 2020). It is also likely that there is continued water usage early and late in the freezing season. In warmer months, when water is more heavily diverted and pumped for irrigated agriculture, observed flows can drop close to zero, while the simulated flow remains high (Fig. 3b). Figure 3c shows a comparison for the same year between the naturalized streamflow and the simulated streamflow. As expected, there is little change in the winter months. However, in the warmer months, we match the magnitude of flows more closely, showing the strong impact of water usage on observed flows.

Figure 3a shows the model comparison to the naturalized streamflow data for the entire period of simulation. We used Spearman's rho as a metric to determine correlation. It tests whether the model is increasing or decreasing at the same time as the observed data and places less weight on the difference in magnitude. In this case, when natural flow is so uncertain, it is more helpful in assessing goodness of fit than the more common Nash–Sutcliffe efficiency. Spearman's rho is 0.72, showing a good positive correlation. However, the model flow still tends to overestimate the peak flows. This is likely due to the uncertainty in our flow naturalization. The naturalization does not account for the impacts of groundwater pumping or water use on other tributaries draining into the Heihe River.

The baseline performance for WTD at 44 observation wells is shown in Fig. 4. The model WTD generally falls within 10 m or less of the observation wells. However, the simulated WTD is significantly shallower where the Heihe River crosses the boundary between the upper and middle basin. This is illustrated by the three dark-blue dots (Fig. 4). This is an area of high $K$ ($\sim 5.6 \, \mathrm{m\,h^{-1}}$), and as a result, without the pumping and diversion that occur here in the managed system, a much greater volume of water can infiltrate at a rapid rate compared to other parts of the domain and raise the model WTD.

There are three additional outlier points where the model WTD is much deeper than expected (orange). The discrepancy at these points is likely to do with our spatial resolution and the uncertainty of the actual well locations in the mod-

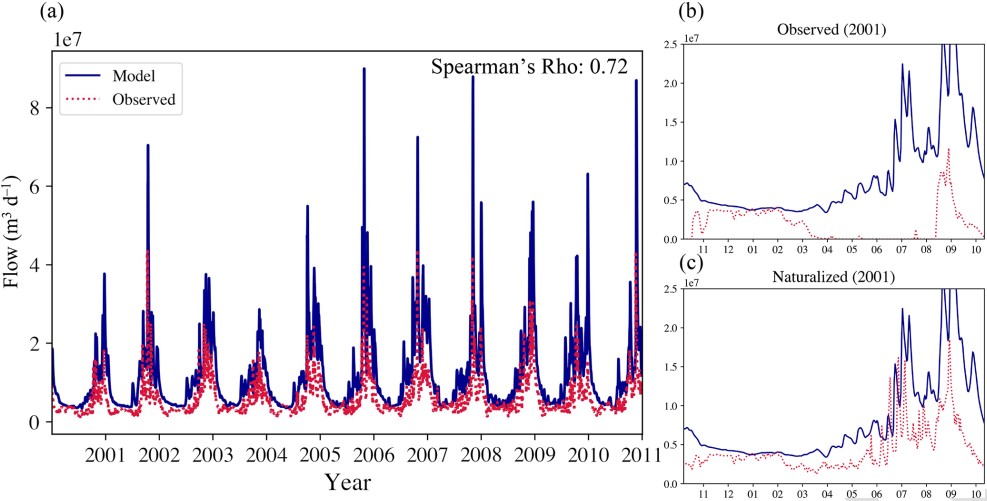

**Figure 3.** A comparison of observed and modeled flow at the HRB2 gage. Panel **(a)** shows the model flow in blue for the entire simulation period compared to naturalized flow data in red for HRB2. Panel **(b)** shows the original observed data in red, which were not subject to naturalization for the 2001 WY. Each number refers to a month, and November 2000 to October 2001 is shown. Panel **(c)** shows the same year but with the naturalized flows in red. Spearman's rho was calculated for the comparison to naturalized flows.

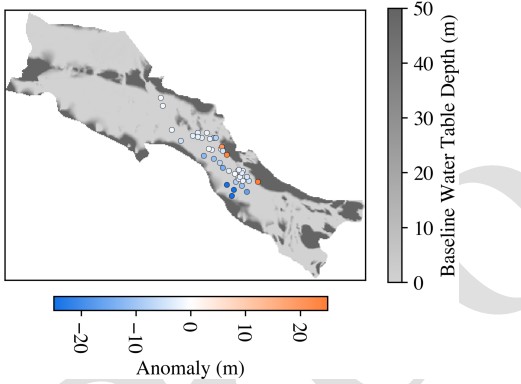

**Figure 4.** The points refer to the locations of the 44 observation wells in the middle basin. They are colored by the difference of the average observed and model WTD for the baseline simulation across the simulation period (2001–2011 WY). A negative value (blue) means the model has a shallower water table, while a positive value (orange) means the model has a deeper water table than observed. A value of zero (white) means there is no difference. The background (gray) shows the mean baseline WTD for the simulation.

eling domain. These wells are located where there is a sharp gradient in WTD due to elevation changes between lowlands and a mountain range in the north. Small differences in well location could result in very different predictions of WTD. Overall, based on the results of both streamflow and WTD, we concluded that the model performs satisfactorily when the natural flow state is considered.

## 3.2 Streamflow

A streamflow time series at the outlet of the Heihe River for each of the five scenarios is shown in Fig. 5. Flows are low in the colder months, consisting almost entirely of baseflow. Most of the streamflow occurs in the late summer and early fall, with high and peaky flows. There is strong overlap between the scenarios, showing that all our climate-warming cases still have consistent overall behavior during simulation. When scenarios do diverge from the baseline, it is typically in the expected order. The permafrost scenario (green) has the highest net increase in flow, followed by the combined (yellow) and warming (orange) scenarios. The glacier scenario (blue) is the only one with flow below the baseline, except for, very occasionally, the warming scenario.

To isolate the scenario impact from the baseline dynamics in the model, we primarily discuss our results in terms of how they differ from the baseline scenario. Inflow perturbation refers to the difference in streamflow input between the scenarios and the baseline. The outlet anomaly refers to the difference in flow at the river outlet. Storage anomaly refers to the difference in storage at a given time. The anomaly fraction (for outlet or storage) refers to the outlet or storage anomaly divided by the inflow perturbation.

Figure 6a and b show the inflow perturbation and outlet anomaly plotted together for the Heihe and Beide rivers. First, in Fig. 6a for the Heihe River, we see that the outlet anomaly is always smaller in magnitude than the inflow perturbation. This means that, as water moves from the inlet to the outlet, the inflow perturbation signal is dampened. That is, a negative inflow perturbation (reduction in water from the baseline) such as in the glacier scenario will become less negative. A positive inflow perturbation (increase in water

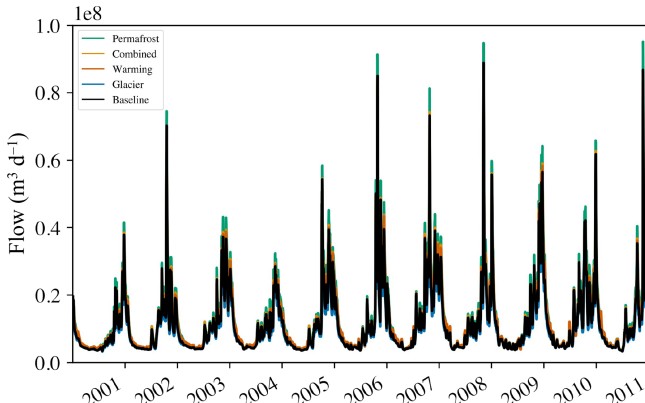

**Figure 5.** Daily streamflow at the Heihe River outlet. Summed from hourly model outputs for the five scenarios: baseline, glacier, permafrost, combined and warming. The scenarios were run with forcing from 2001–2011 WY.

input from the baseline), like the other three scenarios, will become less positive. The only exception is 2011, where the outlet anomaly is slightly less than the inflow perturbation for the permafrost scenario.

5   In Fig. 6b for the Beide River, we see this same dampening signal from 2001 to 2004 for the permafrost and combined scenarios and until 2005 for the glacier scenario. After this year, the inflow perturbation starts to be amplified as opposed to dampened. This is represented in Fig. 6b by 10 the outlet anomaly plotting below the inflow perturbation in the glacier scenario and above the inflow perturbation in the permafrost and combined scenarios. The warming scenario, however, continues to exhibit dampening of the inflow perturbation throughout the simulation.

15   The anomaly fraction refers to the fraction of the inflow perturbation that is still present at the outlet (Fig. 7). A fraction of 1 means that the anomaly at the outlet is equal to the inflow perturbation. A number less than 1 indicates that the outlet anomaly is less than the inflow perturbation and that 20 the signal was dampened. A number greater than 1 indicates the outlet anomaly is more than the inflow perturbation or that the signal was amplified as it moved downstream. If a fraction is negative, it means that the outlet anomaly is the opposite sign of the inflow perturbation – that is, a reduction 25 in flow at the inlet becomes an increase in flow at the outlet or vice versa.

  First in Fig. 7a, we see that the fraction is always less than 1 in all years and all scenarios for the Heihe River, except in 2011 for the permafrost scenario. This corroborates what we 30 see in Fig. 6a, which is that the inflow perturbation is almost always dampened in the Heihe River drainage. The first year of simulation shows a large increase in the fraction for all scenarios, with a smaller increase for the warming scenario. After this point, there is an increasing trend in the fraction for 35 all scenarios from 2001 to 2006. The combined and glacier

scenarios both exhibit a small decrease in fraction from 2003 to 2004. After 2006, trends become much more variable, except for in the glacier scenario, which shows smaller fractional changes from year to year. The warming and combined scenarios show very similar patterns because they have the 40 same inflow perturbation. However, the warming scenario is shifted downwards and maintains a smaller fraction throughout. The warming scenario trends differ from the combined scenario trends in 2010 and 2011. The permafrost scenario shows slightly less variability compared to the combined and 45 warming scenarios. It also exhibits a switch to amplifying behavior in 2011.

  The Beide River shows a large increase in anomaly fraction in the first year in all scenarios except warming, which only has a small increase. This is similar to what is seen in 50 the Heihe River. Likewise, there is a general increasing trend for glacier, permafrost and combined scenarios until 2006. However, after this year, patterns between the two drainages differ. There is a large jump in fraction for the glacier scenario, showing a switch to the amplification of the negative 55 inflow signal. The Permafrost and combined scenarios also exhibit a switch to amplifying behavior, and the fractions are not as variable as for the Heihe River. The differences between the warming and combined scenarios are also more apparent in the Beide than in the Heihe. The warming sce- 60 nario is clearly more variable than the combined scenario in the Beide; however, they are both similarly variable in the Heihe.

  It should be noted that the range of the fractional changes is smaller for the Beide than the Heihe River. The fractions 65 for the Beide range between 0.83 (0.92 without warming) and 1.1, while for the Heihe they range between 0.65 (0.68 without warming) and 1.1 across all years and scenarios. Thus, the overall change in anomaly fraction from the start to end of the simulation period is greater in the Heihe River. 70

  The anomaly fraction is also quantified on a monthly timescale to assess seasonal impacts (Fig. 8). The anomaly fraction cannot be assessed for the glacier scenario in the freezing season because the inflow perturbation is zero from November to March. The anomaly fraction generally in- 75 creases for all scenarios across the thawing season (April to October) as total flows increase. However, the range of months over which the increasing trend persists varies. For the glacier and warming scenarios, it begins in April. The Permafrost scenario has a delay of a few months, with the 80 increase not beginning until June. The combined scenario has the shortest and least consistent increasing window, from July to December. The warming and permafrost scenarios also increase until December, whereas the glacier scenario only increases until October, after which it cannot be as- 85 sessed.

  Variability tends to be consistent between most months across all scenarios. However, there are some months which stand out. These typically correspond to the periods with the lowest flows. For example, the inflow perturbation is small- 90

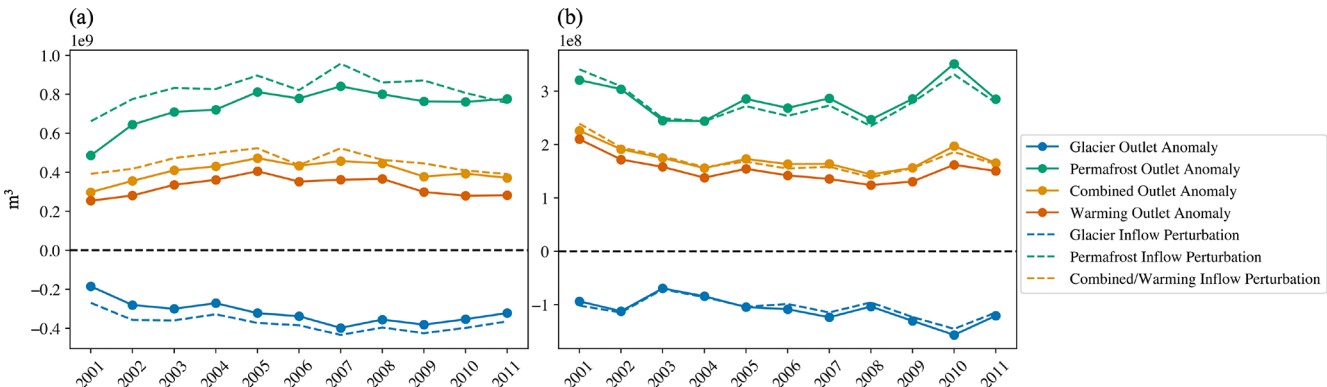

**Figure 6.** Inflow perturbations are shown as dashed lines, representing the magnitude of flow difference between the four scenarios and the baseline. Solid lines show outlet anomalies or differences between scenarios and the baseline at the outlet. Panel **(a)** shows these two metrics for the Heihe River outlet, while panel **(b)** shows the same for the Beide River outlet. The Heihe River is the drainage area for 12 tributaries, while the Beide River is the drainage area for 2 tributaries.

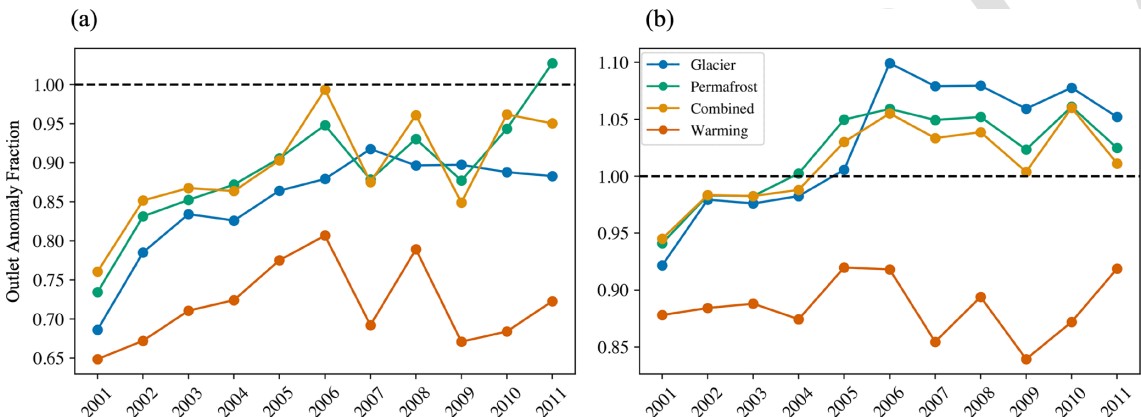

**Figure 7.** Panel **(a)** shows the annual average outlet anomaly fraction (outlet anomaly divided by inflow perturbation) for the Heihe River, and panel **(b)** shows the annual average outlet anomaly fraction for the Beide River for four scenarios (glacier, permafrost, combined and warming).

est (closest to zero) in April for the glacier, combined and warming scenarios. This month is clearly more variable for the combined and warming scenarios but not for the glacier scenario. The permafrost scenario has its smallest inflow perturbation in January, which also corresponds to the month of greatest variability. However, the entire period of December to April has an inflow perturbation of similar magnitude but notably different variability between months.

The anomaly fraction is always positive for the glacier, permafrost and combined scenarios in every month; however, they do differ in the timing of dampening versus amplifying behavior (a fraction less than, as opposed to greater than, 1). Looking at the mean anomaly fraction, the glacier scenario only shows amplifying behavior in October. The permafrost scenario has a mean anomaly fraction above 1 in November and December but shows significant amplifying behavior from September to January. The combined scenario has strong amplifying behavior in April, and the only other

month where the mean fraction goes above 1 is December, with the rest of the year showing dampening. The warming scenario is the only scenario to have a negative fraction at any time of year, with values often going negative in April and May. In these months, even though we added more flow than in the baseline at the inlet, by the time the signal reaches the outlet, the flow is less than in the baseline. Other than these months, the warming scenario shows dampening behavior except for in November, December and March.

### 3.3 Subsurface storage

Subsurface storage has a positive trend in all scenarios, including the baseline (Fig. 9a). This increase in storage in all scenarios is possibly attributed to increasing precipitation in the region. However, major findings are taken with reference to the baseline. This allows us to isolate the processes we are interested in and remove the influence from variables in the climate forcing that are the same across scenarios,

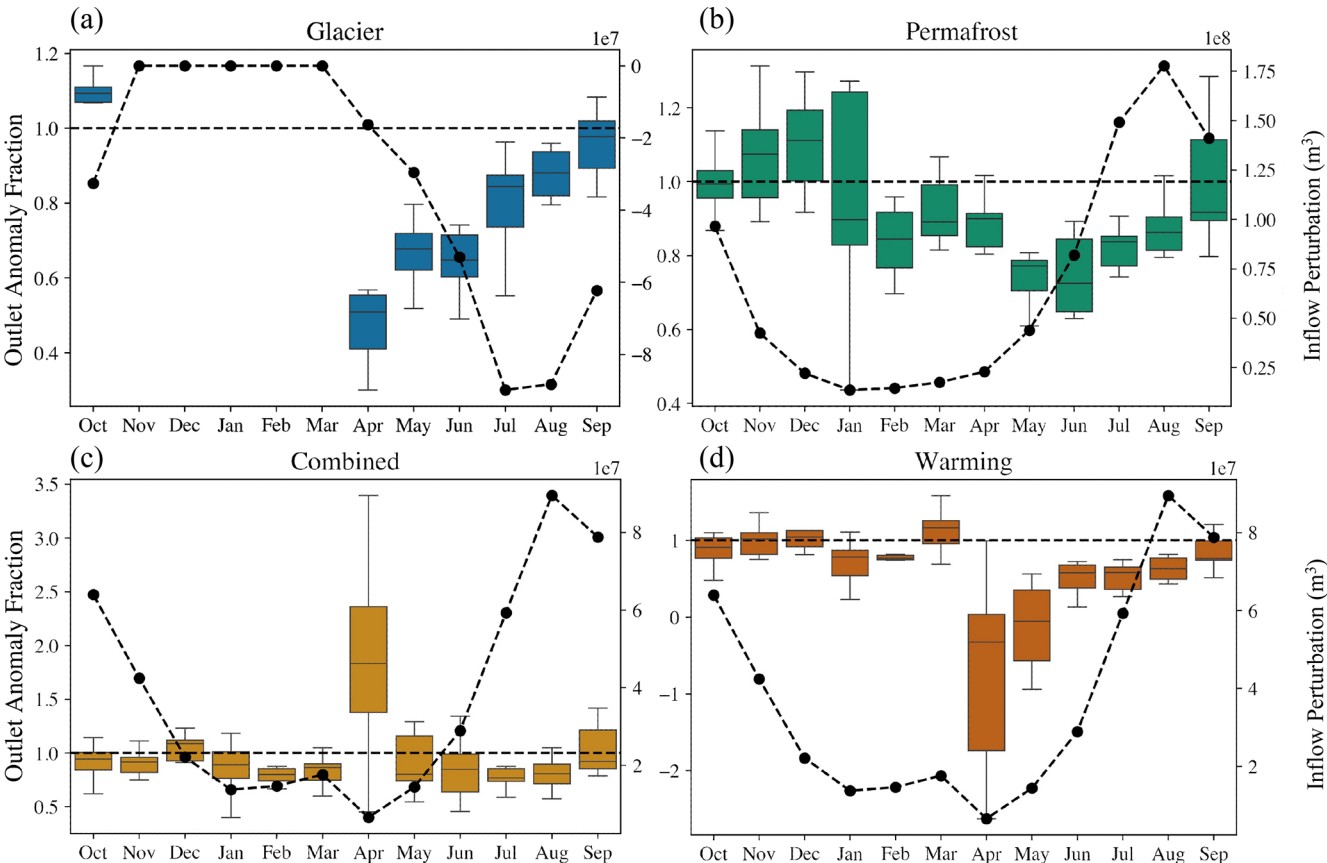

**Figure 8.** A boxplot of the monthly outlet anomaly fraction for the Heihe River outlet for each scenario is shown. Panel **(a)** is the glacier scenario, **(b)** is the permafrost scenario, **(c)** is the combined scenario, and **(d)** is the warming scenario. The colored boxes represent the interquartile range (IQR), and the line in the center is the mean outlet anomaly in that month across all years of simulation. The whiskers extend from ±1.5 IQR and cover 99.3 % of the distribution. The outliers are not shown. Each point on the dashed black line is the average inflow perturbation in that month across the entire simulation period.

such as precipitation. Relative to the baseline, the largest increase in storage is in the permafrost scenario, followed by the combined and warming scenarios, with the glacier scenario losing storage relative to the baseline. The permafrost and combined scenarios have clear positive storage trends over the course of the simulation, except in 2011 for the permafrost scenario. The glacier scenario has a negative trend. The warming scenario has a positive trend until 2005, after which it becomes variable, with no clear trend.

The anomaly fraction for subsurface storage is shown in Fig. 10 for (a) total storage, (b) deep storage (below depth of 10 m) and (c) shallow storage (above depth of 10 m). The glacier scenario has exclusively negative fractions, which indicates a net loss in storage relative to the baseline. The opposite is true for the permafrost and combined scenarios, which both have net gains in storage. In 2011, the negative fraction in the permafrost scenario means that the permafrost scenario did not gain as much storage as the baseline in this year. The warming scenario is variable from year to year, sometimes losing and sometimes gaining storage relative to the baseline.

All anomaly fractions in all simulations in the total (Fig. 10a) and deep subsurface (Fig. 10b) tend to approach zero. This decreasing trend is not as strong in the total subsurface (Fig. 10a), which has a flatter trend. The shallow subsurface anomaly fraction (Fig. 10c) is much more variable, where the fractions tend to approach zero until 2006 and then diverge again. The combination of deep and shallow subsurface trends likely causes the flattening we see in the total subsurface. A computed average of the anomaly fraction in the total subsurface after 2005 results in 0.06 for the glacier scenario, 0.04 for the permafrost scenario, 0.03 for the combined scenario and −0.02 for the warming scenario.

The mean anomaly fraction almost always shows dampening behavior (less than 1) in Fig. 11. This indicates that a quantity of water less than the inflow perturbation applied is added to or lost from groundwater storage. The only exception is the warming scenario (Fig. 11d), with positive amplifying behavior (fraction greater than 1) in April and negative amplifying behavior (fraction less than negative 1) in May. In April, this can be interpreted as a quantity of water

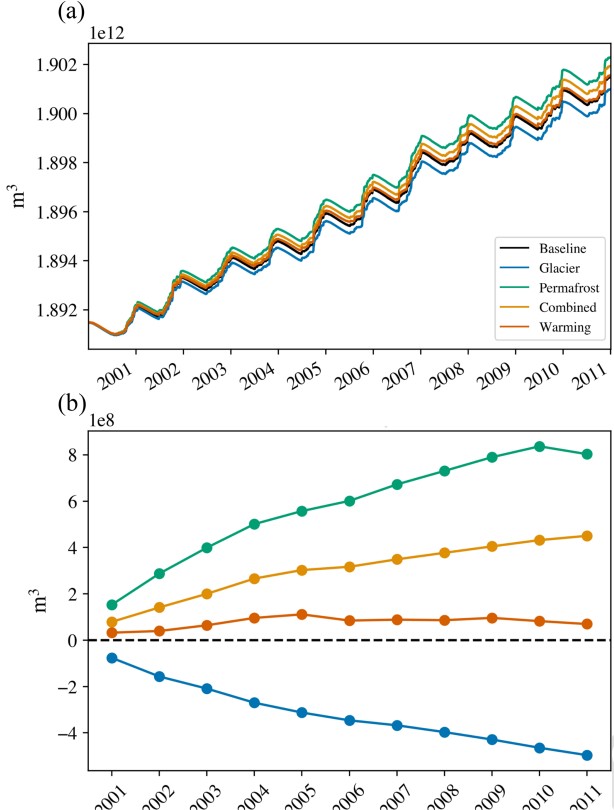

**Figure 9. (a)** Time series of total daily subsurface storage for all scenarios. **(b)** The anomaly of subsurface storage, calculated by subtracting baseline storage from each of the scenarios.

greater than the inflow perturbation being added to groundwater storage. In May, more water is lost from groundwater storage than the inflow perturbation.

When the fraction is opposite to the sign of the inflow perturbation, this represents changes in groundwater storage in the opposite direction to that of the inflow perturbation. In the glacier scenario, this only occurs in April (Fig. 11a). This means that, in April, even though there has been a flow decrease, the monthly storage increase is greater than in the baseline. As for the permafrost scenario (Fig. 11b), while the mean fraction is never negative, there are months across the simulation period that are negative. This means that, even though there is a positive inflow perturbation, there has been a smaller increase in groundwater storage relative to the baseline that month. The combined (Fig. 11c) and warming (Fig. 11d) scenarios both have months where the mean anomaly fraction is negative – May and June for the combined scenario and May through August for the warming scenario.

Where the inflow perturbation is positive, the anomaly fraction increases for all scenarios across most of the thawing season (April to October) as flows increase. Where it is negative, as for the glacier scenario, the trend decreases in a similar time frame. The warming scenario, on the other hand, continues to increase in terms of anomaly fraction until November and exhibits a sharp drop from April to May. In the freezing season (November to March), the combined and permafrost scenarios increase at first in November and December but then begin a decreasing trend until March and April, respectively. The glacier scenario cannot be assessed in the freezing season because the inflow perturbation is zero.

Variability trends are largely consistent between the glacier, permafrost and combined scenarios. The variability tends to be higher in the late thawing season (July to October). The variability tends to be smaller in the freezing season, when flows are lower. The warming scenario is extremely variable in April and May.

### 3.4 The impact of warming

The combined and warming scenarios have the same inflow perturbations at the inlets (Fig. 12). However, the warming scenario has a warming of 2 °C relative to the temperatures in the combined scenario applied across the entire simulation domain. When looking at ET, in the combined scenario in January (Fig. 12a), only small regions of ET are greater relative to the baseline. These regions are only in areas near the inlets. They are also more pronounced in areas which have higher flow, like the main stem of the Heihe.

The warming scenario, on the other hand, shows large differences in ET across the domain relative to the baseline. These differences become more dramatic in July (Fig. 13b), where the maximum ET difference almost quadruples. The differences for the combined scenario are also more pronounced and widespread in July compared to January. It is now possible to see ET differences along the river channel into the domain and not only near the inlets.

The river network relative to the baseline also differs in January versus in July. There is overall greater connectivity of tributaries and higher flows, especially in the combined scenario. Additionally, while the river network between the combined and warming scenarios looks very similar in January (Fig. 12c, d), the river network as it compares to the baseline is noticeably different between them in July (Fig. 13c, d). There is much less flow arriving at both the Heihe and Beide River outlets when comparing the warming and combined scenarios, as well as significantly less connection of inlets to the main river network.

The glacier scenario (Fig. 14a) has an increase in WTD around the river inlets, although this is not as significant in magnitude as the rising water table in the permafrost scenario (Fig. 14b). The combined (Fig. 14c) and warming (Fig. 14d) scenarios show a similar rise in water table near the river inlets. Several inlets have a rise in water table greater than 20 m near the boundary between the upper and middle Heihe relative to the baseline. The areas of greatest increase are not necessarily directly adjacent to this boundary. The warming scenario has broad areas across the domain where the water

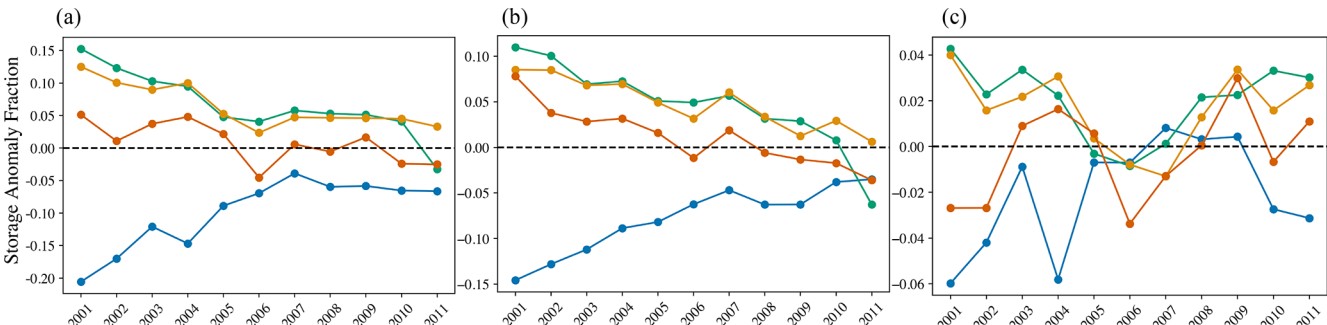

**Figure 10.** The subsurface storage anomaly fraction (storage anomaly divided by inflow perturbation). Panel **(a)** shows the time series for total subsurface storage, while panel **(b)** is for deep subsurface storage, that is depths below 10 m. Panel **(c)** is shallow subsurface storage which corresponds to depths above 10 m.

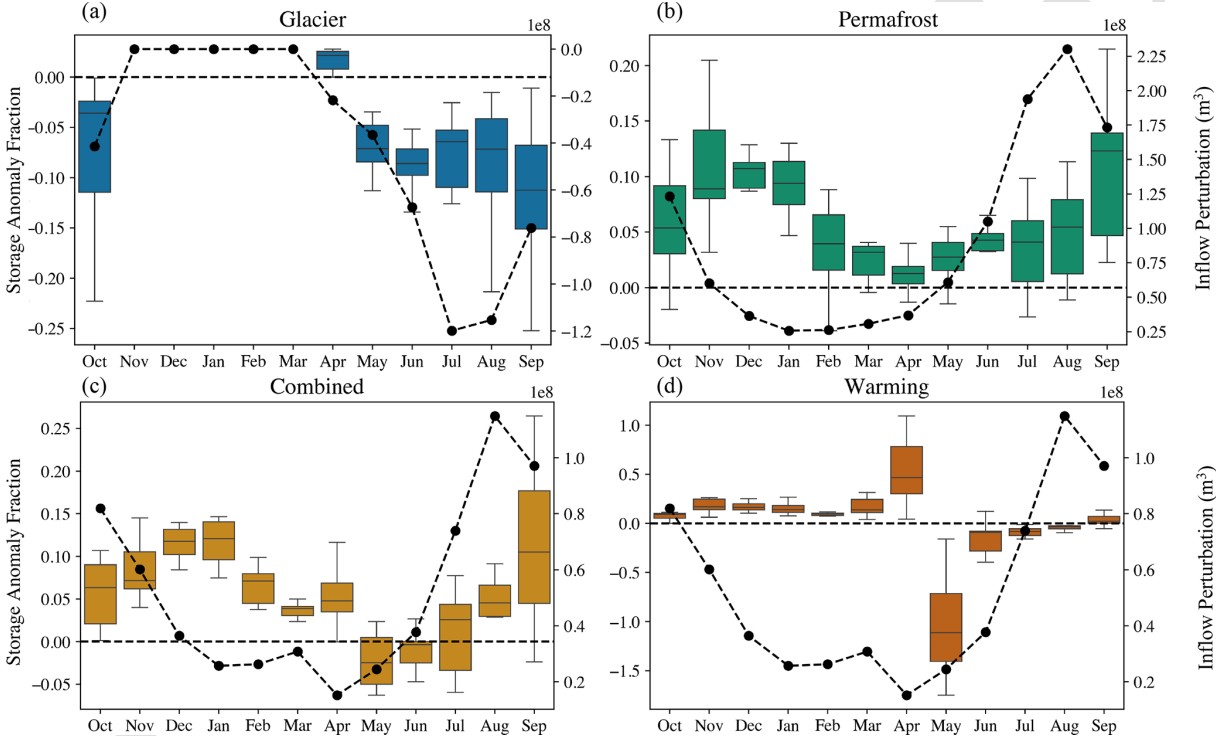

**Figure 11.** A boxplot of the monthly subsurface storage anomaly fraction for each scenario is shown. Panel **(a)** is the glacier scenario, **(b)** is the permafrost scenario, **(c)** is the combined scenario, and **(d)** is the warming scenario. The colored boxes represent the IQR, and the line in the center is the mean storage anomaly in that month across all years of simulation. The whiskers extend from ±1.5 IQR and cover 99.3 % of the distribution. The outliers are not shown. Each point on the dashed black line is the average inflow perturbation in that month across the simulation period.

table falls by several meters. This is not seen in any other scenario and so is almost certainly the result of increasing the temperature.

## 4 Discussion

### 4.1 Mediation of cryosphere-based streamflow changes by the middle basin

First, we will explore the impacts of changes in the upper-basin cryosphere (i.e., the glacier, permafrost and combined scenarios) on the middle basin. The glacier scenario has an overall decrease in streamflow relative to the baseline, while

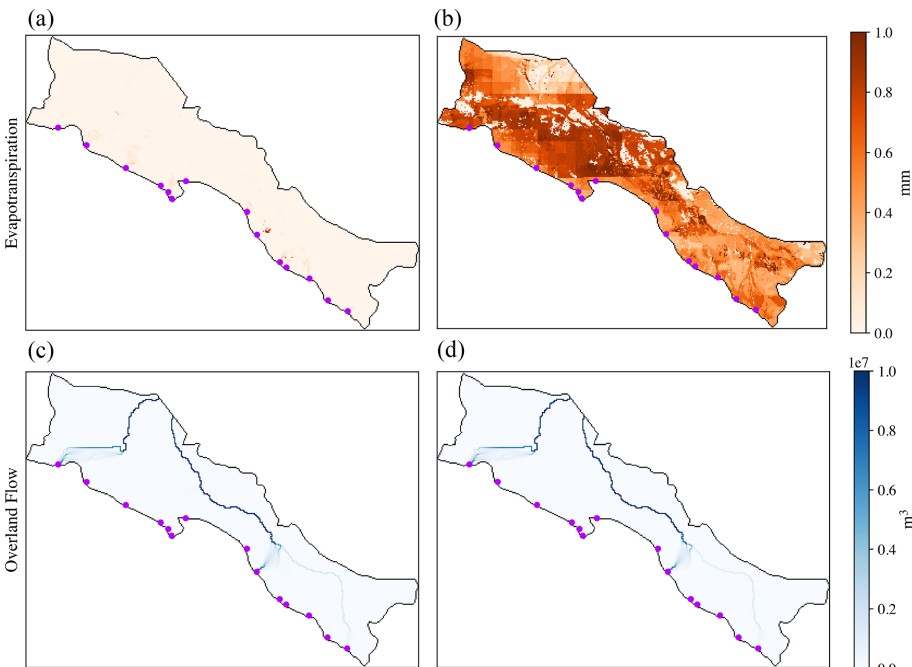

**Figure 12.** The difference of the average monthly sum of ET in January compared to the baseline for the combined scenario **(a)** and the warming scenario **(b)**. The difference in the sum of overland flow in January from compared to the baseline is shown in **(c)** for the combined scenario and **(d)** for the warming scenario.

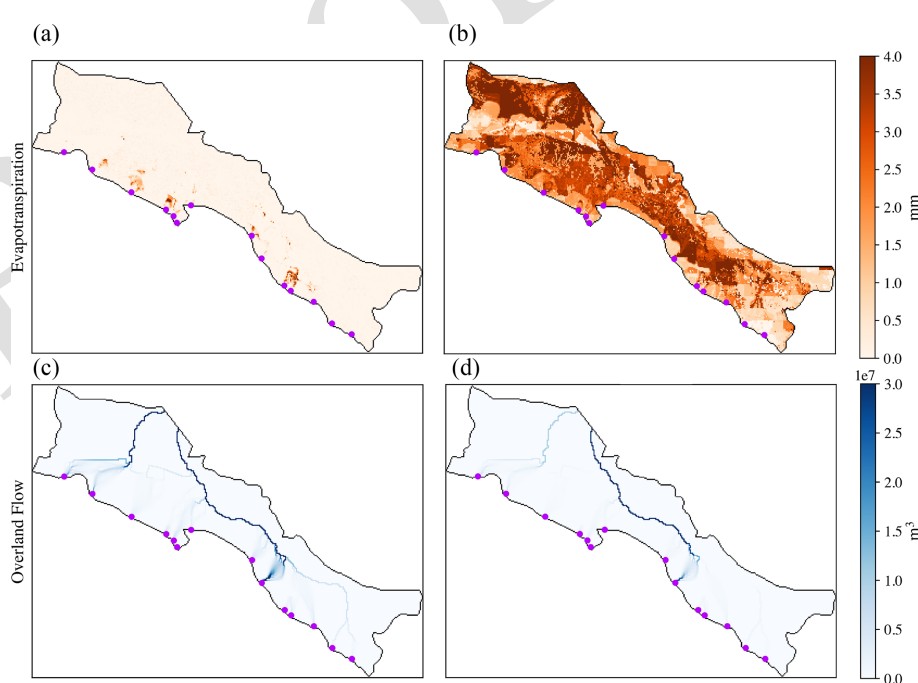

**Figure 13.** The difference of the average monthly sum of ET in July compared to the baseline for the combined scenario **(a)** and warming scenario **(b)**. The difference in the sum of overland flow in July compared to the baseline is shown in **(c)** for the combined scenario and **(d)** for the warming scenario.

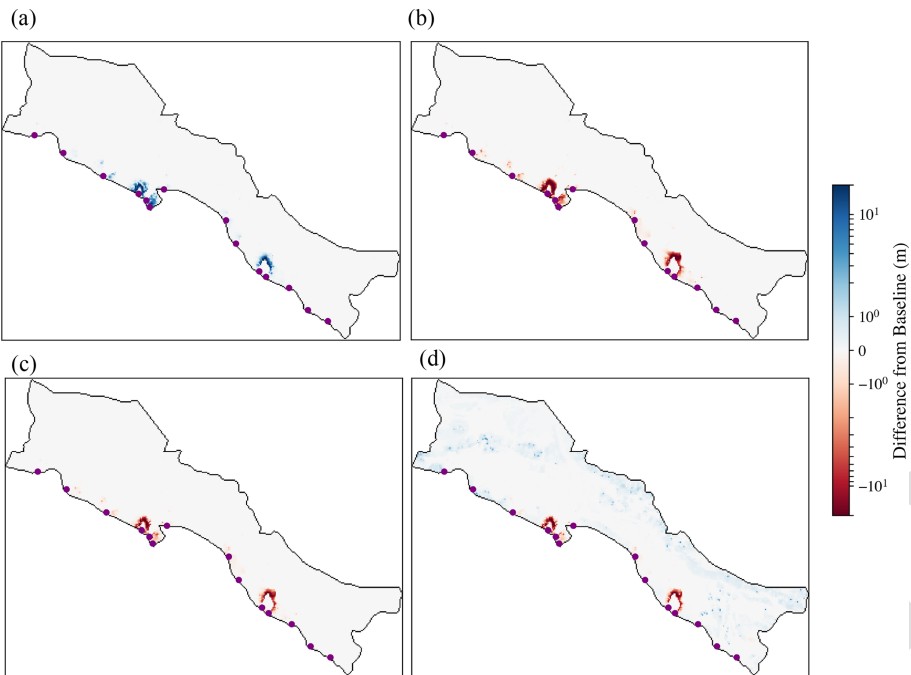

**Figure 14.** The difference in WTD compared to the baseline after 11 years of simulation. Panel **(a)** shows the glacier scenario, **(b)** is the permafrost scenario, **(c)** is the combined scenario, and **(d)** is the warming scenario. The scale is in meters on a log scale, with red being decreases in WTD (rising water table) and blue being increases in WTD (falling water table).

the permafrost scenario has an overall increase. Throughout all scenarios across all years (apart from 2011 for the permafrost scenario), only a fraction of whatever change is applied at the inlet is still present at the outlet (Fig. 7). When streamflow is decreased, stream height falls in the river channel, resulting in increased baseflow from groundwater storage being released to the stream (in locations where the groundwater is shallow and connected). The net result is that streamflow losses are compensated for by groundwater discharge, resulting in a negative inflow perturbation at the inlet becoming less negative as we move downstream to the outlet (Fig. 6). When we increase flow, the opposite occurs (i.e., stream level increases can induce increased groundwater recharge or decrease groundwater discharge). The net effect is a buffering of the streamflow perturbation, in this case the positive inlet perturbation decreasing as you move downstream to the outlet (Fig. 6).

The buffering effect is largest in the first few years, when the water table is at its greatest distance from the baseline scenario. For the glacier scenario, this means that the most water leaves storage to augment streamflow, and for the permafrost scenario, this means that the greatest amount of water is added to storage. As the water table and stream height equilibrate, the gradient decreases, and we expect smaller volumes of water to be exchanged. However, applied differences in streamflow with regards to the baseline result in permanent differences in behavior between the scenarios as a new equilibrium is reached. After 2002, there is little shift

in the rolling average of the anomaly fraction. This indicates that the system has reached a new equilibrium.

On average, after 2002, 88 %, 92 % and 91 % of the applied streamflow perturbation propagates downstream for the glacier, permafrost and combined scenarios, respectively (Fig. 7), while 8 %, 5 % and 5 % of the perturbation is compensated for by changes in groundwater storage (Fig. 10). This leaves 4 %, 3 % and 4 % of the streamflow perturbation, which may be compensated for by a shifting relationship with ET. Figure 13b shows elevated ET for the combined scenario relative to the baseline, particularly near inlets and major river branches, which supports this assessment.

Multiplying the storage anomaly fraction by the average annual inflow perturbation for each scenario allows us to estimate the total annual change in groundwater storage (note that the magnitude of the inflow perturbation varies across scenarios). We obtain a $39.5 \times 10^6 \, \mathrm{m}^3$ decrease of water each year for the glacier scenario, an increase of $56.8 \times 10^6 \, \mathrm{m}^3$ for the permafrost scenario and an increase of $33.5 \times 10^6 \, \mathrm{m}^3$ for the combined scenario. This means that the increase in baseflow from permafrost degradation will more than offset the reduction in flow from glacial loss, as we see in the combined scenario. It should also be noted that we selected an upper-bound scenario for glacial flow reduction. So, it is highly likely that this is a conservative estimate and that subsurface storage will increase in the future in the middle basin as a result of these two process changes.

Estimates of groundwater pumping in the middle basin can range anywhere from $220 \times 10^6$ m$^3$ (Zeng et al., 2012) to $858.6 \times 10^6$ million m$^3$ yr$^{-1}$ (Tian et al., 2018) after adjusting for groundwater being a presumed 30 % of total water usage (Tian et al., 2018) and the middle basin accounting for 90 % of water usage (Liu et al., 2009; Deng and Zhao, 2015). This results in a range of 3.9 %–15.0 % of annual usage added to groundwater storage in the combined scenario depending on the pumping estimate used. This is likely to be highly impactful to human systems because the perturbation is applied to rivers, around which most human settlements and agriculture are located. However, this large uncertainty in groundwater usage estimates makes it difficult to assess the impact of these upstream inflow changes.

Watershed response to streamflow changes varies with the initial state of the water table. In the baseline scenario, groundwater is already shallow across a large portion of the domain, especially near the river network (Fig. 5). The areas of the domain with a deep water table at the start of the simulation tend to be close to the boundary with the upper basin, where there is a larger elevation gradient (Fig. 1). This means that increased flows have a limited area within which to infiltrate and add significantly to storage compared to the baseline. This is seen in Fig. 14a, which shows limited regions with a significant rise in water table. Saturated subsurface conditions combined with a high-precipitation year result in the behavior illustrated in 2011 for the permafrost scenario. Here, despite increased flows, less water is added to storage than in the baseline because a higher fraction of water runs off (Figs. 9b and 10a). It is worth noting then that, over time, permanent increases in relation to subsurface storage may not be as significant as the increase in streamflow would suggest. However, if we consider the system with human impacts of pumping and diversion, it is possible that we would continue to see streamflow supplementing subsurface storage in a more stable way year after year.

## 4.2 Differences between the Heihe and Beide rivers

The Heihe River drains most of the gaged tributaries (12) coming into the basin, while the Beide River in the western part of the domain only drains 2 gaged tributaries before crossing into the lower basin (Fig. 1). The result is a significantly smaller stream network draining to the Beide. Additionally, the distance from the inlet to outlet is, on average, shorter for branches of the Beide than of the Heihe River (Fig. 1). Finally, there is a greater variation in WTD around the tributaries of the Heihe than of the Beide (Fig. 3). These factors combined result in a different groundwater response to streamflow perturbation.

From 2003–2005, the inflow perturbation is always less than the outlet anomaly (dampening) for the Beide River (Fig. 6b), and after 2004, it switches to amplifying behavior (fraction greater than 1) of the inflow perturbation signal (Fig. 7b). The Heihe River, apart from in the 2011 permafrost scenario, always shows dampening behavior (fraction less than 1). In Fig. 14a, the two leftmost gages (purple markers), which drain to the Beide River, have little change in WTD relative to the baseline. This implies that there is little ability for the streamflow signal to be buffered by interactions with groundwater storage. This would explain the lack of significant dampening behavior in the Beide, while for the Heihe, we see dampening that is present but diminishing over time. There are several possible explanations for cases of amplifying behavior. First, in scenarios like the permafrost and combined, which have a net increase in flow, reduced infiltration of that signal over time as shallow groundwater storage fills results in increased runoff. In the glacier scenario, where there is a reduction in flow, an amplification in this negative signal in the Beide may be due to falling water tables (Fig. 14a), which induce further infiltration and streamflow losses (Fig. 7b).

## 4.3 Seasonal differences

The difference in the intra-annual patterns between the scenarios largely depends on whether we are in a baseflow- or runoff-dominated month. In general, the applied streamflow perturbations are most dampened in the early thawing season and summer (i.e., the outlet anomaly fractions are smallest). For example, April is the month with the lowest outlet anomaly fraction for the glacier scenario. In this month, on average, $\sim 50$ % of the streamflow reduction is buffered by either the release of groundwater from storage or reduced ET (Fig. 8a). This is compared to September, where essentially none of the reduction is buffered, with a mean outlet anomaly fraction close to 1 (Fig. 8a). This means that more flow than expected based on the magnitude of the loss of the glacial fraction will arrive at the outlet during the spring and early summer. Conversely, in scenarios where flow is increased relative to the baseline, such as the permafrost and combined scenarios, the month with the smallest mean anomaly fractions occurs a little later (June for the permafrost scenario and July for the combined scenario). Increased infiltration into the subsurface and greater losses to ET result in a larger fraction of the streamflow increase being lost before the basin outlet. In these cases, despite increased flows at the inlets mediated by permafrost changes, there will be less surface water available in the stream network than expected from spring to mid-summer.

To determine if these differences in anomaly fraction are due to changing relationships with groundwater storage, we can look at Fig. 11. First, for the glacier scenario, the only month with a positive fraction is April. In April, less storage is lost relative to the baseline. This is counterintuitive when considering the small outlet anomaly fraction (Fig. 8a). However, the glacier scenario has the same inflow as the baseline during the freezing season, allowing for large increases in groundwater storage. The switch from a positive to negative fraction from April to May signifies that any surplus storage

gained in the freezing season is lost by May. This accounts for the degree of dampening seen in the outlet anomaly fraction in April. For the rest of the thawing season, the glacier scenario gains less storage than the baseline.

In the permafrost and combined scenarios, the storage anomaly tends to increase throughout the thawing season and into the early freezing season (Fig. 11). While the magnitude of the inflow perturbation increases, there is also an increase in the variability and range of the storage anomaly fraction. The variability tends to decrease in the freezing season, while the storage anomaly fraction remains high. Looking at July as an example (Fig. 11b, c), depending on the year, up to 10 % of the inflow perturbation could be added to subsurface storage. However, in other years, that fraction can be negative. That is, despite elevated flow over the baseline, less storage was added for the scenario that year. This is likely to be related to increasing flow across the entire thawing season. If subsurface storage near the stream network is fully saturated, then more of the inflow perturbation will pass through to the outlet and not infiltrate. This is reflected in outlet anomaly fractions approaching 1 across the thawing season (Fig. 8). Regardless of whether there is a decrease or increase in flow, at the end of the thawing season, there is sufficient flow to saturate the subsurface adjacent to the stream network.

Storage anomaly fractions in the freezing season tend to be above zero and less variable (Fig. 11b, c). Lower winter flows result in a smaller likelihood of oversaturating the subsurface near the river network. Second, lower connectivity of tributaries to the main stem during this low-flow period increases the amount of streamflow that infiltrates before arriving at the outlet. This second point can be visualized with differences in river network connectivity between January and July (Figs. 12c and 13c). This means that changes to subsurface storage are more consistent in the freezing season.

To summarize, the results above suggest that, with glacial loss, surface water supply in spring and summer will be more stable than expected due to supplementation by groundwater. However, this means that, in the late summer and fall, groundwater supplies will be more depleted than in the past. As for the permafrost and combined scenarios, a larger fraction of the streamflow perturbation infiltrating in spring and summer means that flows will not be as high in the stream network as would be expected from the magnitude of the flow increase. However, streamflow should be at anticipated levels in the late summer and fall. As for groundwater, the supply should be more robust overall, but the permafrost-mediated changes to baseflow will have a more predictable impact on groundwater storage in the freezing season than in the thawing season.

## 4.4 The influence of warming temperature

Increasing the temperature in the middle basin changes many of the overall impacts of the combined scenario discussed in previous sections. Changes to streamflow impact WTD and

ET in a limited area of the domain (Fig. 14). Warming the domain, on the other hand, will impact the entire middle basin. In Figs. 12b and 13b, ET is elevated across the domain compared to in the baseline. Increases in ET across the domain in the warming scenario result in lower flows on the main river stems and a loss of connection of smaller tributaries in the warming scenario compared to in the combined scenario even though they have identical inflows (Fig. 13c, d). Impacts are less pronounced in January (Fig. 12c, d) when ET is lower.

Increased ET reduces shallow groundwater storage and decreases the chance of oversaturating the subsurface during high-flow summer periods. These two factors combined cause dampening to persist throughout the simulation period for the warming scenario. For example, in Fig. 6a and b, both rivers have consistent dampening behavior in each year of simulation. This is not the case for the combined scenario, which switches to amplifying behavior in the Beide and is variable in the Heihe. Likewise for the outlet anomaly fraction, more than 80 % of the flow increase is never present at the outlet for the Heihe River (Fig. 7a). Less runoff reaching the stream network due to increased ET also contributes to this result.

The warming scenario also has a small net loss in subsurface storage (equating to ~ 0.24 % of annual use) (Figs. 10a and 14d). This is due to diffuse, small drops in WTD throughout the domain. The warming scenario does have a similar rise in water table near the river inlets to that in the combined scenario (Fig. 14c, d), which does not fully counteract the losses in subsurface storage. The warming scenario also has more variable and negative fractions compared to the combined scenario in the shallow subsurface (Fig. 10c). Additionally, less water being available to infiltrate results in a steady declining trend in the deep subsurface over time (Fig. 10b). If we assume that rising WTD and increasing groundwater storage near the inlets slows down over time as indicated, reductions in groundwater storage due to warming may be more significant relative to the baseline than they appear initially.

The warming scenario also shows markedly different behavior than the combined scenario in the spring. In April and May, when streamflow increases at the inlet are small, increases in ET are larger than the streamflow perturbations (Fig. 8d). As a result, the net impact is a streamflow decrease at the outlet relative to the baseline. The warming scenario also gains less storage compared to the baseline throughout the summer, rarely showing increases in subsurface storage relative to the baseline until October (Fig. 11d). This differs from the combined scenario, which shows relative increases in subsurface storage starting in July (Fig. 11c). The behavior between the two scenarios for both the outlet and storage anomalies is similar in the freezing season, where the impact of increased ET, even with 2 °C warming, is minimal. Ultimately, the benefits of a higher-flow regime will not be as

strong in the middle basin in conjunction with the impacts of warming.

## 4.5 Caveats

We have made several simplifying assumptions throughout this research in order to design a well-constrained experiment. However, these assumptions may also influence our findings. We briefly discuss three principal assumptions in our research and how they may impact our results. First, we model a natural flow state even though the middle basin is subject to intensive surface and groundwater usage. Our results are valuable for understanding the physical processes and progression of upstream flow changes in the middle basin. However, these impacts will change when modeled with water usage. Next, we only looked at perturbations to streamflow related to temperature changes in the upper basin. There are other processes that may occur under future climate change which we did not address, such as precipitation changes. However, precipitation trends are less predictable and are difficult to disentangle from the impact of permafrost degradation. Lastly, it would be valuable to run the model for a longer period. This would allow for a better analysis of the long-term response of groundwater storage to changing streamflow. However, we do remain constrained by data availability.

There are two main ways we would like to expand upon this work. First, adding water management operations in the middle basin would give a more realistic view of how these changes will impact the modern basin. While the physical processes are unlikely to change, the magnitude of the impact will shift. Second, it would be ideal to link the middle-basin domain to a model of the upper basin. That way, glacial melt and permafrost degradation would not be simplified and could be linked directly to processes modeled in the middle basin. This would allow for a more physically based change in flow timing and magnitude.

## 5 Conclusion

Climate warming in the upper-basin cryosphere is essentially inevitable. The disappearance of glaciers will decrease overall streamflow, while permafrost degradation will increase baseflow. Examining the downstream impacts on an ecologically and economically important region, such as the middle basin of the Heihe, is of critical importance. Through targeted changes to upper-basin discharge and middle-basin temperature, this study provides valuable insight into the future of water resources in the middle basin. Overall, our results indicate that there will likely be an increase in streamflow and groundwater storage as a result of combined changes to discharge coming out of the cryosphere. Additionally, even when reductions are severe, such as in the glacier scenario, impacts to middle-basin water supply are not as extreme as expected. Groundwater exchanges can mediate some of the short-term impacts and dampen the overall shift in streamflow volume.

These cryosphere-mediated shifts vary seasonally in terms of their influence on the stability and volume of surface and groundwater storage. This is important to understanding the impact the use of surface or groundwater may have at different times of the year for activities such as irrigation. Groundwater supplementation of the glacial fraction makes the stream network more resilient than expected in the spring but at the cost of reduced groundwater storage in the summer and fall. As for the permafrost-mediated changes, they tend to promote increased infiltration in the spring and early summer. This means that, while there may be lower flows throughout the stream network than anticipated by the increases observed at the inlet, the result is larger and more reliable groundwater stores in the middle basin.

However, the impacts of warming on the middle basin may be more dramatic than those resulting from streamflow changes. This is because streamflow impacts tend to be in a limited area around the stream network. We find that widespread warming can overwhelm the signal from streamflow shifts occurring in the upper basin through increasing ET, which thereby reduces streamflow and groundwater storage in the middle basin. Our findings are relevant to other semi-arid basins with mountainous water sources that are facing uncertainty and water stress under climate warming.

*Code and data availability.* All data and programming scripts used to produce this work can be found on CyVerse under the following DOI: https://doi.org/10.25739/kmk7-b046 (Triplett, 2022). Any data or scripts not included in this repository can be made available upon request. Please refer to NASA (2009, https://data.tpdc.ac.cn/zh-hans/data/a78ea495-3db6-40e0-a985-fa845d176fba) and Wang (2014, https://doi.org/10.3972/heihe.093.2014.db) for DEM and land cover datasets respectively.

*Author contributions.* AT, under the advisement of LEC, built the model and designed and ran the scenarios. AT also analyzed all the data and created all the figures with feedback and guidance from LEC. AT wrote the paper with revision and editing provided by LEC.

*Competing interests.* The contact author has declared that neither of the authors has any competing interests.

*Special issue statement.* This article is part of the special issue "Hydrological response to climatic and cryospheric changes in high-mountain regions". It is not associated with a conference.

*Acknowledgements.* The authors would like to thank our collaborators at the Southern University of Science and Technology, particularly Yong Tian, who was integral in providing source data and information about the Heihe River basin. For the use of the supercomputing resources to run the simulations, the National Center for Atmospheric Research is thanked. Finally, our three anonymous reviewers who helped to improve the quality of this manuscript are thanked.

*Financial support.* This research has been supported by the National Science Foundation (grant no. 1855912).

*Review statement.* This paper was edited by Songjun Han and reviewed by three anonymous referees.

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
