# Peer review of "Climate warming-driven changes in the cryosphere and their impact on groundwater—surface water interactions in the Heihe River Basin"

_Hydrology and Earth System Sciences, 2022_

## Author Response (AR1)

This version of the detailed response to reviewers has been updated to what was initially submitted to reflect the actual (as opposed to predicted) changes made to the manuscript. Included line numbers are now with reference to the most recent version of the manuscript.

*Reviewer One:*

*This manuscript analyzed the hydrological response for cryosphere change and temperature increase in the middle Heihe River basin. The topic is interesting. I have some major concerns listed below:*

Thank you for taking the time to provide us with meaningful feedback about our manuscript. We have provided detailed responses to each of your concerns below. We hope that the explanations and suggested revisions to address concerns are satisfactory.

1. *About the PF-CLM model simulation. Because the interaction between the surface and subsurface flow is important, the flow in the river channel in the study region need to be accurately simulated. So, what method is used for the river routing in the study region?*

ParFlow is a fully integrated ground and surface water model, it links surface and subsurface flow with an overland flow boundary condition. When ponded water occurs on the surface, ParFlow uses the kinematic wave approximation to solve for overland flow. With this approach there is no need to determine a priori if a cell is a river cell or not, or to have a separate river routing model. In ParFlow, streams can form and disappear over the course of the simulation as areas become ponded or dry out. A full description of this coupled approach is available in a publication by Kollet and Maxwell from 2006, "Integrated surface-groundwater flow modeling: a free-surface overland flow boundary condition in a parallel groundwater flow model." We updated Lines 115-116 to make it clear that a reader should look here specifically for details on the overland flow formulation.

2. *The pumping of groundwater for irrigation may have significant impact on groundwater simulation in the study region. How is this considered in the model simulation.*

Pumping groundwater assuredly has an impact on both ground and surface water in the Heihe River Basin. We added a few more targeted references to address this (Lines 199-200, 204-205). However, we chose to model a natural flow state in order to isolate the impacts of a warmer climate on streamflow and groundwater storage. Our goal is to quantify how much warming could change the overall water supply of the basin (i.e., the water that managers would have available to them). If we had included management operations simultaneously, they could overwhelm the signal we are trying to see. We have added this justification supported by the reference Terrier et al. (2020) which provides an in-depth overview of natural flow state modeling. We agree that that groundwater pumping is important to the region though and needs to be treated carefully.

In response to this comment, we added a new section 2.4 Natural flow state modeling to more clearly explain the decision to model a natural flow state and the possible impacts of that decision as well as clarifying this decision and its impacts at the start of section 3.1 Baseline model performance.

3. *For the glacier scenario, a 15% percent of flow is considered as the contribution of glacier melt. Is this fraction provided by previous observations or other studies?*

We have clarified the estimates from our referenced studies which use a variety of methods to estimate glacial contribution to streamflow on Lines 278-279. To expand on our decision to use 15% as our glacial fraction, we decided to reduce it by the largest amount we considered possible, higher than the largest

literature estimates for the Heihe River of around 10% (Chen, 2014; He et al., 2008; Yang, 1991) for three reasons: First, estimates in the literature of glacial contribution to streamflow are based on historic melt rates and cryosphere interactions and, as such, cannot account for unforeseen nonlinearity under future climate change. Second, it is challenging to gather fine resolution data in mountainous environments such as the Qilian mountains, adding uncertainty to the assessment of the glacial fraction. Finally, many studies are for the mainstem of the Heihe River, and while it is the largest contributor of water to the downstream basin, other tributaries do contribute to the total basin water balance and may have smaller or larger fractions of glacial contribution (Li et al., 2014).

We preferred to err on the side of overestimating the overall glacial fraction so that our findings can be used as an outer bound of what might be expected. The goal of the study is to look at possible directions and rough magnitudes of trends driven by a warming cryosphere (as opposed to giving exact predictions of what the basin will look like in 2050 for example). Thus, we decided that setting a lower bound on future water supply would give the most information while maintaining a low total number of runs. In response to this comment, we added a shorter version of this explanation on Lines 279-284.

4. *The groundwater storage increases in all the scenarios, this may be related to the precipitation changes. So, the results of all the scenarios are also related to the precipitation. Then the conclusions are also need to be revised and analyze the precipitation conditions is need. I suggest to add some scenarios, such as a scenario without the increase in precipitation.*

It is likely that precipitation changes as a result of climate change are occurring in the basin and some studies do indicate there is an increasing precipitation trend in the region (Shi et al., 2006; Zhang et al., 2016). However, we decided explicitly to exclude modeling precipitation changes because of greater uncertainty and variability in future precipitation trends, especially in high mountain regions. Additionally, by adding more processes to track, it becomes more difficult to isolate the impact of a few changes, namely cryosphere melt and temperature, which this study was focused on.

We would also like to note that our results are not strongly impacted by this trend because all the scenarios were subject to the same precipitation forcing data. All of our major results are taken with reference to the Baseline, eliminating the influence of precipitation on storage trends. In response to this comment, we updated Section 3.3 Subsurface storage (Lines 438-441) to explain that increasing storage could be due to increasing precipitation trends and explain why we think that is not impactful in this study.

5. *Figure 3. It seems that the model overestimated the observed flow, why?*

Yes, the model overestimates flow, particularly peak flows. This is primarily to do with the fact we are modeling a natural flow state, while the observed data we must compare to is subject to operations. This means that in spring and summer months, when water is diverted to reservoirs and for irrigation, our model does not capture that diversion. Additionally, groundwater pumping is not included. Excluding these processes impact how our modeled flow compares to observed. We adjusted the observed streamflow timeseries according to Zhang, A. et al. (2015) to account for surface diversion, which improved our match. However, we should note that this adjustment doesn't account for groundwater pumping or long-term differences in water table and how they could influence peak flow trends. Our winter month baseflow, when minimal operations occur, was a good match to observations though and we determined that the model is behaving reasonably for our intended natural flow conditions. As we are modeling a natural flow state, we simply have more water than observed data and should expect our flows to be higher. The authors added a section 2.4 Natural flow state modeling and updated 3.1 baseline model performance to address this point.

***6. For the permafrost degradation, it may reduce the summer peaks, is this effect considered in the study?***

We did not reduce peak flows to explicitly account for possible impacts by permafrost degradation. We did this for several reasons. First, we based our perturbations on previous research in the basin. Gao et al. (2018) found the impact of permafrost degradation in the upper Heihe on streamflow to be a 50% increase in baseflow, which was assessed in winter months due to minimal other contributions to streamflow (Gao et al., 2018). While there are likely reductions to peak flows in higher flow spring and summer months due to permafrost degradation, there are many more contributing sources to streamflow at that time, so it is difficult to attribute what changes are due to permafrost degradation and which are due to other processes, specifically precipitation. Any selection of a reduction of peak flows would be arbitrary on our part since we did not explicitly model these processes in the upper basin and there is no previous research to support the estimates we would need. Attributing changes in peak flows to changes in permafrost coverage would be best answered by a physically based model of the upper basin and is unfortunately outside of the scope of our middle basin model.

In addition to the uncertainty noted above, we would like to point out that due to the steep elevation gradient between the upper and middle basin there is significant groundwater recharge from the mountain front happening upstream of our model boundary. Thus, we expect that some of the changes the reviewer notes here may already be accounted for in the baseflow adjustment, which may have been greater without reduced peak flows upstream.

Finally, we would like to highlight that similar to the glacial reductions to streamflow, we are attempting to set outer boundaries to the water supply of the naturalized basin. By not reducing the upper basin peaks, we get the upper boundary that may be available to water managers in the future, while the glacier scenario provides us with the lower boundary.

The authors have updated Section 2.6 Cryosphere melt scenarios to address this comment (Lines 1481-1492.

***Reviewer Two:***

*This study constructs numerical models to simulate the hydrological processes in the Middle Heihe River Basin in response to glacier loss, permafrost degradation, and temperature increase. The topic is interesting and the study area is an important area. However, there are some problems in the manuscript, which are listed as following.*

We thank the reviewer for their detailed comments to improve this manuscript and have replied to all major concerns below. We hope that the explanations and planned revisions to address questions and concerns are satisfactory.

Major comments:

1. *Commonly, specific yield is smaller than porosity. In the manuscript, the authors used specific yield as an analogy for porosity. In addition, the authors simplified specific yield values in the model. Specific yield data from 17 unique values calibrated in Tian et al. (2015a) were simplified to three intervals of 0.1, 0.2, and 0.3. The authors stated that this simplification was used to lessen computational demand. The reviewer doubts the reasonability of such simplification. This*

*oversimplification may cause the model far from real condition and the results may be inaccurate. As supercomputer and parallel computation is so common, computation demand may not be a problem.*

These are great points of clarification that should be made in the updated manuscript. First, the authors acknowledge that specific yield is not the same as porosity. In this study, we used the same source data as that for the previously published HEIFLOW model. The HEIFLOW model only contains data for specific yield, while the ParFlow model requires porosity as an input. We decided the specific yield values could be used as reasonable estimates of porosity for two reasons. First, the middle Heihe is mostly an unconfined aquifer, with discontinuous aquitard sections (Yao et al., 2015a). This was confirmed with the input data. In unconfined aquifers, specific yield can more feasibly be used as an analog for porosity, although there is certainly some difference between the two values. Second, we note that the underlying data used to build the specific yield estimates was sparse and uncertain. We make the assumption here that the differences between porosity and specific yield are smaller than the uncertainty of the specific yield values themselves which is explained in more detail below.

As stated above, we initialized our model with the same input data used in the HEIFLOW model (Tian et al., 2015b) and with data provided from the Heihe Program Data Management Center. However, we found that using many of the values directly caused unrealistic behavior in our model, not only for specific yield but for hydraulic conductivity. Furthermore, we also found some of the values from the raw data to be non-physically realistic (i.e. anisotropy values of 8000). Given these complications we determined that we would need to conduct a separate model calibration exercise for our simulations. In the initial testing process, we found that our results were more sensitive to hydraulic conductivity than porosity, so we focused our efforts on calibrating that variable. By taking a lumped approach, we acknowledge how uncertain these variables are, and focus on more impactful hydrogeologic variables like K in our system.

Furthermore, we would like to note that when we say to lessen computational demand, we mean the computational demand to perform satisfactory calibration on all possible input variables. This will be clarified in the future manuscript. We absolutely do take advantage of parallel computing in our simulations, and ParFlow has been demonstrated in previous work to have excellent parallel scaling performance. However, it should be noted, that compute resources are still limited by compute allocations and must be used responsibly. One year of simulation with ParFlow alone (i.e., for our spin-up runs) requires roughly 250 core hours (running on 972 cores for about 15 minutes). One year of ParFlow-coupled to CLM requires roughly 4800 core hours (running on 972 cores for about 5 hours). Considering the number of years for the groundwater system to return to equilibrium, especially after changes to hydrogeologic variables, this is a significant computing constraint.

The authors have updated Section 2.3 Model inputs, Lines 147-155 to address this comment.

2. *The vertical thickness of the model is 472 m, which may be too shallow for a groundwater model. Many regional groundwater models have vertical thickness of 3-5 km. Can the 472 m thickness capture the main groundwater flow system in the study area?*

The previous modeling studies of the Heihe are our best source for the hydrogeology of the basin. As explained in our previous answer, we did have to calibrate our model and change the individual values in the domain, but we tried to maintain the general lithology that they developed. The maximum depth of hydrogeologic data used in the previous modeling efforts was 2094m (Yao et al., 2015a; Tian et al., 2015b) with a no-flow boundary imposed below this. After analyzing the data, only a very small percentage of the domain had data below 1072m (about 8%). Additionally, only about 50% of the MHRB

domain has data at depth greater than 472m and a K greater than .005 m/h. Originally, we had an additional bottom layer that was 600m thick in our model to cover the depth of the previous models. Note that the HEIFLOW model has variable thickness while our ParFlow model has constant thickness.

Our study is focused on shallow groundwater and groundwater surface water interactions, and we expected that the deep regional flow paths extending below this depth would have a limited impact on our results especially given the other uncertainties involved. We decided to test without the last large thickness layer. After comparing our results before and after this change, we did not see large enough differences in water table depth and streamflow processes on the time order of our simulations (11 years) to warrant the additional computational costs of keeping it.

The authors have updated Section 2.5 Model configuration and initialization, Lines 221-228 to address this comment.

3. *For a regional groundwater model with relatively large thickness, the decrease of hydraulic conductivity K and specific storage Ss with depth should be taken into account. Did the authors consider the decrease of K and Ss with depth in their model?*

We used hydraulic conductivity and specific storage input data as referenced in Tian et al. 2015b and used in additional studies with the HEIFLOW model (Tian et al., 2018; Sun et al., 2018). The creation of these datasets based on observations is described in Yao et al. (2015a). In this dataset, K is shown to decrease with depth. Additionally, any calibration we did was scalar, so the depth relationship we observe in the underlying hydraulic conductivity data has been maintained. The specific storage data, which we used unaltered from the source, was uniform with depth. We believe that some of these discrepancies in the data, for instance only two specific storage values that do not vary with depth, versus 92 values for K which do, highlights the limitations in comprehensive data for regional scale models.

The authors have updated Section 2.3 Model inputs, Lines 150 and 156-157 to address this comment.

4. *The constant flux boundary condition along the border between the Upper and Middle Heihe may not be reasonable. The flux from the Upper Heihe is variable, and the flux is different in different seasons and among different years. The authors should clarify why a constant flux boundary is reasonable.*

The reviewer is correct that we would expect groundwater flux to change over the course of the year, however our ability to represent this is limited by observations and we don't have any data to support seasonal variations on the boundary. Tian et al. (2015b) also used a constant flux value based on average annual data, literature estimates and model calibration (table 1) which we translated to our model and further details can be found in that publication. Furthermore, we do think the constant flux boundary is defensible for the following reasons: First, the groundwater flux represents only a small fraction of total water input to the domain, about 5% on average. Thus, the expected range of seasonal flux around this value is not expected to significantly impact the total amount of water entering the domain. We also performed calibration on the groundwater boundary condition. The values we tested varied between +/- 75% of the original data. The change showed a minimal impact on groundwater and surface water. Seasonal flux changes seem unlikely to fall outside of that range.

Further, there is a large elevation gradient between the upper and middle basins in the Heihe, so a large fraction of groundwater discharges as streamflow at this boundary which is the same boundary the constant flux boundary condition is applied. So, seasonal changes in baseflow variability will be captured by the surface water inputs to the model.

The authors have updated Section 2.5, Lines 229-238 to address this comment.

5. *The authors stated that "about 75% of water coming into the middle basin domain is from streamflow, 20% from precipitation, and 5% from the groundwater boundary condition." Where are the percentages from? Are there any evidence for these percentages?*

We calculated the annual average volume of streamflow input into the domain from historic time series (Table 1, Heihe Program Data Management Center). We then calculated average long-term recharge or the balance of precipitation (Xiong and Yan 2013) with ET (PML V2 ET product, National Tibetan Plateau Third Pole Environment Data Center - http://data.tpdc.ac.cn/en/data/48c16a8d-d307-4973-abab-972e9449627c/?q=PML_V2), to obtain the rough contribution of precipitation to the water budget. Last, we obtained the magnitude of water entering through the groundwater boundary condition as calibrated in Tian et al. (2015b). The resulting breakdown of these water sources is approximately 75%, 20% and 5%.

The authors have update Section 2.5, Lines 242-245 to address this comment and clarify the above point.

6. *A uniform water table depth of 20 m was used in the model. Why the authors use a uniform water table depth? Is there any support material, publications, or evidence for such a water table depth? For such an area, water table depth should be different in different regions.*

We did not use a uniform water table depth for simulation. We started from a uniform water table depth of 20m before spin-up in order to have a point to initialize from. We then proceeded to run for 115 years to get a new water table to use in the calibration process. At this point, we have a spatially variably water table depth and we deemed the spin-up to be complete because storage as a change of percent recharge was less than 1%. After this, we calibrated the model and after final parameters were selected, we ran for an additional 73 years, again until storage change as a percent of recharge was less than 1%. At this point, we have the water table from which we initialized our simulations.

The authors deleted the confusing line about starting from a 20 m uniform water table depth before either spin-up or calibration. We deemed this line to be confusing and also unnecessary detail as it had no impact on any model running (all models need to initialize from some state). We also clarified the language on Lines 246-254 to address any remaining confusion.

*For the Combined scenario, the authors used 15% reduction in thawing season flow and 50% reduction in baseflow. Are these reduction percentages have any supporting data, references, or other evidences? Or are they chosen arbitrarily?*

We understand from this comment and others that the source of the values for the simulations were not sufficiently clear in the manuscript. We give a brief explanation here of where the values come from and

will improve the clarity of this explanation in the manuscript. Additionally, we refer the reviewer to our response to reviewer 1 in comment 3 where we address a similar point.

First, to address the 15% glacial reduction. We did consult the literature to assess these values, and the language has been clarified on Lines 278-279. Estimates in the literature of glacial contribution to streamflow are based on historic melt rates and cryosphere interactions and cannot account for unforeseen nonlinearity under future climate change. Additionally, many studies are for the mainstem of the Heihe River, and while it is the largest contributor of water to the downstream basin, other tributaries do contribute to the total basin water balance and may have smaller or larger fractions of glacial contribution (Li et al., 2014). Thus, we decided to reduce the glacial fraction by the largest amount we considered possible which is higher than the largest literature estimates for the Heihe River of around 10% (Chen, 2014; He et al., 2008; Yang, 1991) to account for this. We could then conclude that any other reasonable fraction used would result in a smaller impact and use that to guide the conclusions at which we arrived. The goal of the study was to look at possible directions and magnitudes of trends and not give exact predictions of what the basin will look like in 2050 for example, so we considered the 15% value selection the one that would give the most information. This has been clarified and addressed in Section 2.6, Lines 279-284.

For the permafrost scenario, we used a study by Gao et al. (2018) which presented data for the increase in winter baseflow in the upper Heihe. Gao et al. (2018) state that since there are no (or few) other contributions to streamflow other than groundwater discharge in this period that any increase to winter flows would be directly caused by increases in baseflow from permafrost degradation. By linear interpolation of their data, the increase over a 30-year period of winter baseflow was 50% for an 8% loss in permafrost area. Assuming a similar loss in permafrost area in an additional 30 years, we increased baseflow by another 50%. Even though the impact of permafrost degradation to streamflow can most easily be assessed in winter, the changes to hydraulic conductivity caused by permafrost degradation are permanent. Thus, we assume the baseflow impact applies year-round. We have clarified our logic for the selection of 50% within Lines 285-296.

Finally, we added Lines 308-309 to ensure the reader knows the changes for the Combined scenario follow the same logic as above.

7. *For comparison of the observed and modeled flow at the HRB2 gage, it can be seen from Figure 3 that the modeled values are significantly smaller than the observed values. The fit between observed and modeled data should be greatly improved.*

The authors are in agreement that the modeled flows (dark blue, figure 3) are significantly higher than the observed flows (red, figure 3). The authors were modeling a natural flow state, while the data we had to calibrate the model to were subject to operations. Operations include reservoir and canal diversions as well as groundwater pumping which are significant in this region. For this reason, we do not expect to be able to match the observed flows. In fact, if we are modeling the natural system correctly, it is quite impossible that we would match the observed data in such a heavily managed system. For this reason, we chose to focus on matching winter baseflows when there are few water management operations occurring and also chose to naturalize the data to better assess our fit. After those adjustments, our streamflow comparisons are more reasonable (note that we also discuss this point in our response to reviewer 1 in comment 5). The authors added Section 2.4 Natural flow state modeling and updated the start of Section 3.1 Baseline model performance to reflect the answer to this and other related comments.

8. *The authors use Spearman's rho as the standard to determine correlation. Why not use the Nash–Sutcliffe efficiency coefficient to evaluate the fit between observed and modeled data? The Nash–Sutcliffe efficiency coefficient is a widely accepted standard for this purpose.*

To also address the reviewers concerns about using Spearman's Rho instead of Nash-Sutcliffe efficiency, we wanted to be able to assess improvements in our fit from calibration that were not so heavily weighted by missing high magnitude peak flows in spring and summer as we expected to not fit those well due to modeling a natural flow state. Spearman's Rho tells us if we are getting directional changes right, with less emphasis on magnitude. Thus, it is a more appropriate metric for us to compare our natural flow state to observations subject to operations.

This all being said, we understand how this seemingly very poor fit would erode confidence in the findings of the study. So, we will add a section highlighting (1) the fact that we are modeling a natural flow state, (2) justifying that decision, (3) better highlight our comparisons to the baseflow and naturalized streamflow which are a more reasonable metric for comparison here, and (4) speaking to why that still allows this study to make the conclusions it does. This was carried out in Section 2.4 Natural flow state modeling and the start of Section 3.1 Baseline model performance. We specifically address the point about Nash–Sutcliffe efficiency in Lines 340-343.

Minor comments:

All minor comments will be fixed as specified below in the final manuscript. Extra care will be taken to review all references and assure that they have the correct family and given name both in the manuscript and reference section so that credit is properly attributed. Some references in within the manuscript and within the reference section were updated to reflect this. Further, an additional detailed review for grammar and linguistic clarity will be done before resubmission. Many small edits were made to the manuscript, but the authors were careful to ensure they did not change the intended meaning of the manuscript and were only made to correct errors, typos, clarity, grammar and to fit HESS formatting standards.

1. Add a space between numbers and its units, throughout the manuscript.
2. Line 20: 2C should be 2$^{\circ}$C, there are other places in the manuscript
3. Line 40: Zongxing et al. should be Li et al., Li is the family name. Please correct this throughout the manuscript.
4. Line 85: "ground and surface water" should be "groundwater and surface water"
5. Lines 216 and 231: Abbreviation "2011WY" should be defined at the first time appearance.
6. Line 494: There is a typo: 3.32.9
7. Figure 14: There is no need to give two color legends.
8. Check the correctness of the references. For example, In Line 801: Zongxing L. should be Li, Z., Li is the family name and Zongxing is the given name. Similar mistakes lake Hongyi L. and Yongge L.

---

## Author Response (AR2)

*Reviewer One:*
*The author explained most of the comments of reviewers. I feel that there is still a problem. The whole country boundary of China and Taiwan province of China should be shown in Figure 1 (including country boundary of China on the ocean)*

We thank the reviewer for their feedback and are happy to hear our previous responses were acceptable. The authors' intention with the inset in Figure 1 is to provide a geographical context for readers who may be unfamiliar with the location of the Heihe River Basin on a global scale. To improve on this with the reviewer's feedback in mind, a new map has been selected which expands on the surrounding geographic area and includes the areas noted above. In addition to the new map, the text in the figure description has been updated to make the authors' intention clear.

*Reviewer 2:*
*The manuscript aims to predict the impact of cryosphere changes on groundwater-surface water interactions in the Heihe River Basin. Overall, this manuscript has been improved, and is more valuable. However, I think it's better to provide more details about the model initialization and conclusion. Specific suggestions are as follows:*

We thank the reviewer for their suggestions and appreciate their positive review. The additional minor revision comments were very helpful in continuing to improve the clarity and scientific relevance of the manuscript. All minor comments were addressed and fixed in the manuscript text. Full responses to the specific suggestions are below.

1. *Lines 249-253. About the starting point of the model for all scenarios, authors mentioned "…… a two-year spin up coupled to CLM using climate forcings from the 2011 WY was performed. ……ran for 55 years with ParFlow only. Then, for an additional 18 years coupled to CLM using the 2001 WY and 2002 WY climate forcings." Why the two-year spin up coupled to CLM using climate forcings from the 2011 WY? Moreover, why choose "55 years" and "18 years" to ran the model rather than other number of years, such as 50 years and 20 years? How many years did the model run from the starting point for all scenarios? Did the model only run 11 years, i.e., from 2001 WY to 2011WY for all scenarios?*

It was our intention to provide as much transparency as possible about the spin-up process, however in this case we think the details we provided are obscuring the main point which is only that we successively initialized our model to achieve a stable equilibrium condition before running simulations (as is common practice in watershed modeling). The simulation lengths for the spin-up steps are determined entirely by the performance of the model and cannot be predicted which is why the numbers seem random. We have decided in response to this comment to simplify the main text to remove the unnecessary detail that was provided in the original manuscript. In the revised text, we now focus on how we determined when spin-up was complete rather than describing all individual year lengths since that is the more important factor.

However, all values will be clarified in detail here. First, all of our year lengths correspond to the point at which we achieved steady state when we spun-up the model at various stages and could

not be pre-determined. In every stage of spin-up we evaluated the equilibrium conditions of the model by comparing changes in storage to other water balance components. It took 115 years to bring the ParFlow model into steady state. We chose 2011 as that was our calibration year to ensure we would not see large fluxes in the pressure field when we began calibration coupled to the climate model. It is not in this version of the text, but we also did an abbreviated spin-up of the model after each parameter change because it takes time for the groundwater system to move and to see the real impact of the parameter changes made.

After we got our best set of parameters from calibration, another spin-up was necessary to make sure that the scenarios with the new parameter set would start from a steady state. 55 years is how long it took to achieve steady state with the long-term recharge forcing, and 18 years running with the coupled climate model to bring it as close as possible to what the groundwater system would look like at the start of the scenarios (2001WY). In fact, we surpassed our target of storage change as a percent of recharge of 1 % in both these cases out of an abundance of caution. It cannot harm the results to spin-up for longer, the only cost is increased compute time. After we had this new pressure state, that is what we used as a starting point for all scenarios. All running before this point was only to get a properly calibrated and initialized model to run the scenarios from. Each scenario was then run from this starting point for 11 years each for the reasons given from Lines 282-286.

2. *I suggest a summary of the effects of cryosphere changes on seasonal hydrological characteristics to consistent with the objective of this paper stated in Lines 88-91.*

We appreciate the suggestion and have updated the text in both the seasonal discussion (Section 4.3) and the conclusion to more closely reflect the objectives stated on Lines 88-91. This includes specific statements on the impacts to surface and groundwater storage depending on the season for each of the scenarios.

Minor comments:
1. *The number of section headings from section 3.2 to section 3.5 is incorrectly marked.*
This has been corrected
2. *Missing units in right ordinate in figure 8 and figure 11.*
This has been corrected
3. *Does "15.0-3.9%" refers to "15-39%" in Line 539?*
It is supposed to be 3.9–15.0 % and has been corrected.

In addition to the specific minor comments above, some grammatical changes were made to the text none of which changed the meaning. Some of these changes were in figures, but only to assure that mathematical units were in the form expected from HESS and nothing in the actual scientific content of the figures.